# PRSformer: Disease Prediction from Million-Scale Individual Genotypes

**Payam Dibaeinia[1]**
p.dibaeinia@gmail.com

**Chris German[1]**
chrisg@23andme.com

**Suyash Shringarpure[1]**
suyashss@gmail.com

**Adam Auton[1]**
aauton@23andme.com

**Aly A. Khan[1,2]***
aakhan@uchicago.edu

[1]23andMe, Palo Alto, CA, USA
[2]University of Chicago, Chicago, IL, USA

## Abstract

Predicting disease risk from DNA presents an unprecedented emerging challenge as biobanks approach population scale sizes ($N > 10^6$ individuals) with ultra-high-dimensional features ($L > 10^5$ genotypes). Current methods, often linear and reliant on summary statistics, fail to capture complex genetic interactions and discard valuable individual-level information. We introduce **PRSformer**, a scalable deep learning architecture designed for end-to-end, multitask disease prediction directly from million-scale individual genotypes. PRSformer employs neighborhood attention, achieving linear $O(L)$ complexity per layer, making Transformers tractable for genome-scale inputs. Crucially, PRSformer utilizes a stacking of these efficient attention layers, progressively increasing the effective receptive field to model local dependencies (e.g., within linkage disequilibrium blocks) before integrating information across wider genomic regions. This design, tailored for genomics, allows PRSformer to learn complex, potentially non-linear and long-range interactions directly from raw genotypes. We demonstrate PRSformer's effectiveness using a unique large private cohort ($N \approx 5M$) for predicting 18 autoimmune and inflammatory conditions using $L \approx 140k$ variants. PRSformer significantly outperforms highly optimized linear models trained on the *same individual-level data* and state-of-the-art summary-statistic-based methods (LDPred2) derived from the *same cohort*, quantifying the benefits of non-linear modeling and multitask learning at scale. Furthermore, experiments reveal that the advantage of non-linearity emerges primarily at large sample sizes ($N > 1M$), and that a multi-ancestry trained model improves generalization, establishing PRSformer as a new framework for deep learning in population-scale genomics.

## 1 Introduction

Learning predictive models from high-dimensional, complex structured data is a fundamental machine learning challenge. This challenge is acutely relevant in modern genomics, where biobanks are rapidly scaling towards **million-sample sizes** ($N > 1,000,000$) and individual genomes are characterized by hundreds of thousands to millions of genetic variants (e.g., Single Nucleotide Polymorphisms, SNPs), yielding a regime of **ultra-high dimensionality** ($L > 100,000$). Effectively leveraging

---

*Corresponding author.

39th Conference on Neural Information Processing Systems (NeurIPS 2025).

individual-level genomic data at this unprecedented $N \times L$ scale is critical for unlocking deeper insights into complex trait genetics, such as predicting disease susceptibility [1, 2].

Current state-of-the-art methods for disease risk prediction primarily rely on Polygenic Risk Scores (PRS) derived from Genome-Wide Association Study (GWAS) summary statistics [3, 4, 5, 6]. While effective to a degree, these methods are predominantly linear, capturing additive genetic effects. Furthermore, by operating on summary statistics, they discard potentially valuable individual-level information and struggle to model non-additive genetic interactions (epistasis) [7, 8, 9, 10]. These limitations may cap predictive performance, especially as dataset sizes scale towards the million-sample regime where subtle interaction effects might become detectable.

Transformer architectures have revolutionized sequence modeling in other domains by capturing complex, long-range dependencies via self-attention [11]. We hypothesize that the attention mechanism provides a powerful inductive bias for genomics, enabling more effective modeling of pairwise linear and non-linear interactions between genetic loci compared to traditional architectures or inherently linear models. However, a critical barrier persists: the prohibitive $\mathcal{O}(L^2)$ computational complexity of standard self-attention renders its direct application to genome-scale sequences ($L > 100,000$) computationally infeasible. Addressing this scalability bottleneck is critical to harnessing the power of Transformers for large-scale genomics.

Here we introduce **PRSformer**, a novel Transformer-based architecture specifically engineered for scalable, end-to-end, multitask disease risk prediction directly from ultra-high-dimensional ($L > 100$k) individual-level genotypes ($N > 1$M). PRSformer's core innovation lies in its scalability, achieved by incorporating neighborhood attention (NA) [12], an efficient attention mechanism restricting computations to local genomic windows, resulting in $\mathcal{O}(L)$ complexity per layer. This design aligns with the biological structure of the genome, which we treat as a series of linked regions, called linkage disequilibrium (LD) blocks, where genetic variants are often inherited together. PRSformer stacks NA layers, which first model interactions within LD blocks, then progressively integrates information between neighboring LD blocks in deeper layers, capturing larger-scale genetic patterns influencing disease.

To evaluate PRSformer's ability to harness the shared genetics underlying multiple, related traits, we trained and validated it in a multitask setting across 18 autoimmune and inflammatory conditions. This trait set provides an ideal testbed for multitask learning due to immune-mediated inflammatory diseases frequently exhibiting shared inflammatory pathophysiology and overlapping genetic factors [13]. The multitask formulation allows us to test PRSformer's ability to exploit shared genetic associations while learning disease-specific patterns, aiming to enhance predictive performance through a shared representation.

We provide rigorous empirical validation using data from a unique large private cohort ($N \approx 3.8$M European-ancestry individuals) for training, validation, and evaluation across D = 18 autoimmune and inflammatory conditions using $L \approx 140$k variants. The scale of this cohort significantly exceeds current public biobanks [14], providing a critical real-world testbed for methods designed to handle genomic data of the million-scale magnitude. We conduct stringent comparisons against: (i) highly optimized linear models (regularized logistic regression with learnable embeddings) trained on the *exact same individual-level genotype data*, isolating the benefit of PRSformer's non-linear architecture; and (ii) state-of-the-art summary-statistic-based PRS methods (LDPred2 [5]) derived from the *exact same cohort*, enabling a direct comparison between end-to-end and summary-statistic-based approaches. Our experiments demonstrate statistically significant performance gains for PRSformer.

Our main contributions are:

- **Scalable deep learning for genomics at population scale:** We present an efficient multitask Transformer architecture applied to population-scale data ($N \approx 5$M) with ultra-high-dimensional features ($L \approx 140$K), establishing a blueprint for tackling other genome sequence prediction tasks.

- **Critical scaling law for non-linear models:** We empirically establish and quantify a key scaling law demonstrating that the predictive advantage of non-linear models over linear methods emerges primarily at the million-sample scale for complex immune-related conditions. Our analysis quantifies this effect, showing that performance gains grow consistently as the training set size increases beyond one million individuals.

- **Multitask learning improves genomic prediction:** We show that multitask training across related traits consistently outperforms the standard single-task paradigm, demonstrating the benefit of learning a shared genetic representation across complex immune-mediated inflammatory diseases.

- **Improved cross-ancestry generalization:** We show that training PRSformer on multi-ancestry data, including an additional $\sim 1.1$M non-European individuals, markedly improves prediction accuracy for held-out non-European individuals compared to a model trained only on European-ancestry data, offering a path toward more equitable genomic prediction.

## 2  Related work

This work is situated at the intersection of statistical genetics, genomics, and deep learning. We specifically advance upon prior work in three key areas: polygenic risk prediction, the application of deep learning to genomic data, and the development of efficient Transformer architectures for ultra-long sequences.

### 2.1  Polygenic risk score methods

Traditional complex trait prediction relies heavily on PRS derived from GWAS summary statistics [4, 6]. Early methods often involved simple thresholding and summing of SNP effects [3, 15]. More recent Bayesian approaches, such as LDpred2 [5] and PRS-CS [16], explicitly model LD patterns and utilize shrinkage priors to improve predictive accuracy. These methods represent the current state-of-the-art for prediction from summary statistics. However, PRS methods based on summary statistics are fundamentally limited in several ways.

First, by discarding individual-level genotype and haplotype information, these methods cannot capture LD structure and must instead rely on LD estimates that are typically imputed from external reference panels, which can introduce biases due to population mismatches [17, 18]. Second, they cannot capture variant-variant interactions such as epistasis as they are restricted to using marginal variant effects. Third, the use of precomputed summary statistics constrains these models to largely linear architectures, precluding the discovery of complex multi-locus or hierarchical genetic patterns. Recent summary-statistic approaches to leverage non-additive signal remain constrained by the lack of individual-level haplotype context [19]. Taken together, these limitations may cap predictive performance, particularly as biobank-scale datasets grow large enough to enable the detection of more subtle and nonlinear genetic effects.

Alternative individual-level approaches, such as BOLT-LMM [20] and GEMMA [21], estimate SNP effect sizes under a linear mixed model framework to account for population structure and polygenic background effects. However, these methods are computationally demanding at our study's scale ($N$=3.8M, $D$=18 traits): GEMMA's cubic complexity in $N$ renders it intractable, while BOLT-LMM, though more scalable, operates on a single trait at a time, requiring 18 separate runs. Prior work has shown that LDPred2 achieves predictive performance comparable to BOLT-LMM across multiple traits [22, 23], supporting its use as a strong linear baseline for comparison.

### 2.2  Deep learning in genomics and trait prediction

Deep learning has been successfully applied to various supervised genomic prediction tasks. Much work has focused on modeling sequence-level information (DNA base pairs) to predict molecular phenotypes like transcription factor binding [24, 25, 26], chromatin accessibility [27], or gene expression [28, 29]. These approaches have predominantly utilized Convolutional Neural Networks (CNNs) or Transformers incorporating CNN-style tokenization, which are well-suited to capturing biologically meaningful motifs and local patterns at base-pair resolution. However, this paradigm is less intuitive when modeling the influence of genetic variants (e.g., SNPs) on complex traits, as causal variants can be spread across the genome and may interact over long distances, often without strong local sequence motifs defining their impact.

The application of deep learning to predict complex traits (like disease status) directly from *individual-level genotype data* (i.e., SNP arrays) remains relatively underexplored, particularly at the population scale addressed in this paper [30, 31]. This is largely due to the challenges of ultra-high dimensionality

($L$) and, until recently, the limited statistical power of publicly available cohorts with both individual-level genotypes and phenotypes ($N$). Prior work in this specific domain has often relied on: (i) tree ensemble models such as gradient boosting or simple neural networks trained on reduced feature sets (e.g., using LASSO feature selection); (ii) smaller cohorts where complex interactions are difficult to detect; and (iii) models operating on precomputed PRS or summary statistics rather than raw genotype data [8, 32, 33, 34, 35, 36]. Recently, Phenformer [37] proposed a multi-scale Transformer that predicts disease risk from DNA sequences by linking genetic variation, gene expression, and phenotype through a pretrained sequence-to-expression backbone. While conceptually similar to our end-to-end genotype-to-phenotype goal, Phenformer operates on DNA sequences covering approximately $\approx 3\%$ of the genome and is trained on $\sim 150\text{K}$ individuals, whereas our approach models variant-level genotype data and scales to millions of individuals, enabling systematic analysis of how nonlinearity interacts with data scale in complex trait prediction.

### 2.3 Efficient transformer architectures

Applying standard Transformers to genome-scale data ($L > 100k$) is computationally prohibitive due to the $\mathcal{O}(L^2)$ complexity of self-attention [11, 38]. A wide range of efficient attention mechanisms have been proposed to address this limitation, including sparse attention patterns (e.g., Longformer [39], BigBird[40]), low-rank approximations (e.g., Linformer [41]), and kernel-based methods (e.g., Performer [42]). Other architectures exploit locality through sliding windows or blockwise mechanisms (e.g., Swin Transformer [43]) to reduce complexity while capturing local dependencies. In our work, we adopt neighborhood attention [12], a variant of self-attention in which each query attends only to a fixed-size local window of neighboring tokens, rather than the full sequence. This inductive bias aligns well with the block-like correlation structure of genomic data driven LD. By limiting attention to a neighborhood of size $k \ll L$, NA reduces both computational and memory complexity to $\mathcal{O}(L \cdot k)$ -achieving linear scaling in sequence length. We employ the optimized GPU implementation provided by the NATTEN library [12, 44], which supports scalable training on long sequences while maintaining the expressiveness of content-based attention.

## 3 Methods

### 3.1 Problem definition

We aim to predict susceptibility to multiple ($D = 18$) autoimmune and inflammatory conditions from individual-level genotypes. Formally, given a dataset of $N$ individuals, the input for individual $i$ is their genotype profile $\mathbf{x}_i \in \{0, 1, 2, \text{UNKN}\}^L$, representing genotypes of $L$ pre-selected genetic variants, where UNKN indicates missing data. The target output is a vector $\mathbf{y}_i \in \{0, 1, \text{UNKN}\}^D$, representing the binary status (case/control) for $D$ diseases, where UNKN indicates unrecorded status. Our goal is to learn a multitask function $f : \mathbb{Z}^L \to [0, 1]^D$ that predicts the probability of each disease $\hat{\mathbf{y}}_i = f(\mathbf{x}_i)$. The primary challenges lie in the ultra-high dimensionality ($L \approx 140\text{k}$ in this work), the need to capture potentially non-linear and long-range interactions, and leveraging the statistical power of population-scale datasets ($N \approx 5\text{M}$ total used in this study). We propose a Transformer-based architecture adapted for this task, leveraging efficient attention mechanisms for scalability and multi-task learning for joint prediction across diseases.

### 3.2 PRSformer architecture

PRSformer adapts the Transformer architecture for disease prediction from ultra-long ($L \approx 140\text{k}$) individual-level genotype sequences using the following key designs:

**Scalability via Neighborhood Attention:** Standard $\mathcal{O}(L^2)$ self-attention is computationally infeasible. We replace it with Neighborhood Attention (NA) [12], restricting each query token's attention to a symmetric local window of size $k$. This reduces complexity to $\mathcal{O}(L \cdot k)$, enabling efficient processing of the $L = 137,245$ input variants used in this study. We use $k = 385$, chosen via hyperparameter tuning (Section 3.5, Supplementary Table F8), which conceptually aligns with capturing dependencies within local LD blocks (Figure 1A) and corresponds to roughly $\approx 100$ kilobases along the genome [45].

**Genome-ordered input without explicit positional encodings:** Input variants are ordered by their chromosomal position and then concatenated from Chr1 to Chr22. This fixed order is used for all

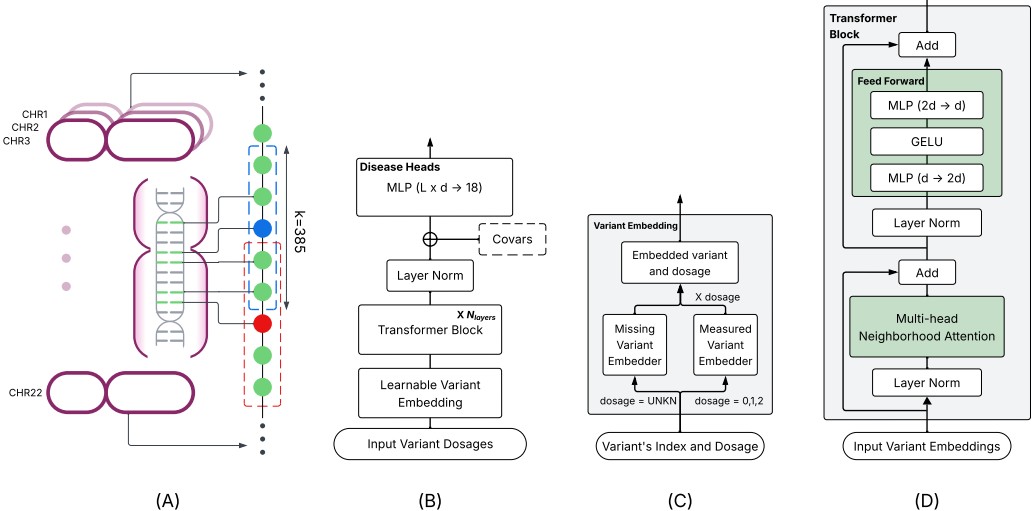

Figure 1: Schematic overview of PRSformer and its core components. (A) A preselected set of variants across the 22 chromosomes, sorted by genomic position, allows each query variant (e.g., blue or red) to attend within a local block ($k = 385$) via neighborhood attention. (B) Model architecture with input, Transformer blocks, and output layer. (C) Variant embedding layer, which encodes each variant and its corresponding genotype (0, 1, 2, or UNKN) into a 64-dimensional representation. (D) Transformer block with pre-layer normalization, neighborhood attention, and GELU activation.

individuals. We omit standard positional encodings (e.g., sinusoidal or learned absolute). The fixed genomic order provides implicit relative positional information that NA inherently leverages within its local attention windows. We also experimented with learned positional encodings, but did not observe measurable improvement in performance.

The overall data flow of PRSformer proceeds as follows (Figure 1B):

1. **Learnable variant embedding layer (Figure 1C):** Each variant in the input sequence of variant genotypes is mapped to a $d_{\mathrm{model}}$-dimensional vector. For each variant $j$ and individual $i$, an observed genotype $x_{ij} \in \{0, 1, 2\}$ is represented as $E_j \cdot x_{ij}$ using a learned variant-specific embedding $E_j$. A missing genotype ($x_{ij} = $ UNKN) is represented by a separate learned variant-specific embedding $M_j$. This allows the model to distinguish missingness from observed genotypes distinctly for each variant. We use $d_{\mathrm{model}} = 64$.

2. **Transformer blocks (Figure 1D):** The embedded sequence is processed by $N_{\mathrm{layers}} = 2$ Transformer blocks. Each block applies pre-layer normalization [46] followed by multi-head NA ($N_{\mathrm{heads}} = 4$) and a feed-forward network with GELU activation [47]. The choice of $N_{\mathrm{layers}} = 2$ was based on validation performance, where deeper models did not show significant improvement for this task (Supplementary Table F7), suggesting two stacked NA layers provide a sufficient receptive field to capture interactions between adjacent LD blocks.

3. **Output layer:** The normalized sequence representation from the last block is flattened and optionally concatenated with covariates such as sex and age, and passed to a fully-connected layer generating $D = 18$ independent disease likelihood predictions. We also evaluated mean pooling and dedicated [CLS] tokens as alternatives to simple flattening of normalized representations, but neither outperformed the flattening-based design (see Supplementary Table F10).

Key architectural hyperparameters ($d_{\mathrm{model}}$, $N_{\mathrm{heads}}$, $N_{\mathrm{layers}}$, NA window size $k$) were optimized based on validation set performance (Supplementary Tables F5-10).

### 3.3 Datasets

We utilized data from a large, private biobank consisting of individuals who consented to participate in research under an IRB-approved protocol. Starting from an internal GWAS data freeze timestamped 08-2021 (used to prevent information leakage, see Section 3.4), we identified individuals genotyped on the same platform and excluded all pairs of individuals related by more than 700 cM (i.e., first cousins or closer), thereby minimizing the risk of learning simple familial signals. Individuals included had recorded phenotypes (i.e., self-reported status) for at least one of the $D = 18$ autoimmune and inflammatory conditions considered. We also excluded individuals under age 10 who did not have a case diagnosis. This resulted in a training set of $N_{\text{train}} = 3,838,549$ individuals of European genetic ancestry (throughout this work, ancestry was determined via an internal classifier [48]).

We constructed temporally distinct validation and test sets using individuals who enrolled and consented after the 08-2021 data freeze date and up to 12-2024, applying the identical filtering criteria. The validation dataset was used for hyperparameter tuning, while the test dataset was used to report final performance metrics. This yielded $N_{\text{val}} = 525,448$ and $N_{\text{test}} = 494,265$ individuals. To assess whether models capture familial relationships versus causal genotype–phenotype associations, we constructed a kinship-controlled European test set ($N \approx 148k$) by subsetting test individuals related to any training sample by no more than 300 cM and ensuring that no pair within the subset is related by more than 700 cM. In total, the European dataset comprised $N = 4,858,302$ individuals across training, validation, and test sets, with an additional $N \approx 1.1M$ non-European individuals included for cross-ancestry training (see Section 3.7). Case/control counts per disease and further details regarding cohort construction (including differences from the subset used for GWAS computations) are provided in Appendix A and Supplementary Tables F2-4.

### 3.4 Variant selection

To define the input feature space ($L$), we selected variants associated with at least one of the $D = 18$ diseases based on internal GWAS summary statistics (European ancestry cohort, computed prior to an 08-2021 data freeze to prevent information leakage into model training). For each disease, variants passing standard QC, located on autosomal chromosomes, had a genotyping rate $\geq 0.95$, MAF $> 0.001$, and exhibited nominal association with the disease (GWAS p-value $< 1 \times 10^{-2}$). The final PRSformer input set was the union across all 18 diseases, resulting in $L = 137,245$ variants. Further details on GWAS procedures, exploration of variant's pruning by LD and per-disease variant counts are in Appendix B and Supplementary Table F9)

### 3.5 Training

Given the training data $\mathcal{D} = \{(x_i, y_i)\}_{i=1}^{N_{\text{train}}}$, we trained PRSformer by minimizing the following loss, summed over individuals and their available (non-UNKN) disease labels $t \in T(i)$:

$$\mathcal{L} = -\sum_{i=1}^{N_{\text{train}}} \sum_{t \in T(i)} [y_{i,t} \log(\hat{y}_{i,t}) + (1 - y_{i,t}) \log(1 - \hat{y}_{i,t})]$$

where $T(i)$ denotes the set of recorded disease statuses for individual $i$. We also evaluated focal loss [49], task-uncertainty–weighted loss [50], and standard averaged cross-entropy, all of which were outperformed by the proposed loss function in terms of validation AUROC. We used the AdamW optimizer [51] ($\beta_1 = 0.9, \beta_2 = 0.999$, weight decay=0.05) with an initial learning rate of $5 \times 10^{-4}$, decreased via a Cosine Annealing scheduler. Training was performed efficiently on this large-scale dataset for 2 epochs consisting of $\approx 120,000$ gradient updates in an effective batch size of 64 across four NVIDIA A100 GPUs, leveraging Distributed Data Parallel and mixed-precision (FP16) training (training duration was tuned based on validation AUROC across diseases (Table F5); most models showed signs of overfitting beyond two epochs). Hyperparameters, including architecture choices ($N_{\text{layers}}, d_{\text{model}}, N_{\text{heads}}, k$), were selected based on optimal AUROC on the validation set after extensive searches (e.g., See Supplementary Tables F5-10), following standard ML best practices to minimize overfitting to validation data and ensure that test performance provides an unbiased estimate of generalization. To isolate the contribution of genotype to model performance, all models presented in the main text were trained without including covariates such as sex or age.

## 3.6 Baseline models

To comprehensively evaluate PRSformer and validate our main claims regarding the utility of end-to-end non-linear modeling on large-scale individual-level genotypes, we established three rigorous baselines. These baselines are specifically designed to: (1) compare against the current state-of-the-art using conventional summary-statistic inputs (LDPred2), (2) benchmark against an enhanced version of this state-of-the-art (Stacked LDPred2), and (3) isolate the specific performance gains attributable to PRSformer's non-linear Transformer architecture via a carefully matched linear counterpart.

1. **LDPred2: state-of-the-art summary-statistic method.** We selected LDPred2 [5] due to its strong empirical performance and widespread adoption in the field [36, 52]. To ensure the most direct comparison possible, we configured LDPred2 meticulously:

   - **Matched data source:** LDPred2 was applied to GWAS summary statistics derived from the *same European training data freeze* used for PRSformer's data and variant selection.
   - **Cohort-specific LD:** An LD reference panel from our research cohort was used.
   - **Standard QC:** Input variants ($\sim$445K per disease) passed standard GWAS QC and LDPred2-specific filtering [18].
   - **Tuning:** LDPred2 hyperparameters ($p, h^2$) were extensively tuned (up to 100 models per disease) by maximizing AUROC on the *same validation set* used for PRSformer (details in Appendix C).

2. **Stacked LDPred2: enhanced summary-statistic baseline.** To create a stronger summary-statistic baseline, we ensembled the converged LDPred2 models from the hyperparameter search using elastic-net regression trained via cross-validation on the validation set (Appendix C).

   *PRSformer+:* Since Stacked LDPred2 uses the validation set for training ensemble weights, we develop and compare it against **PRSformer+**, which is the final PRSformer model retrained on the combined training and validation datasets, ensuring parity in total data usage (Supplementary Figure E2).

3. **Linear model: direct architectural ablation.** This crucial baseline isolates the contribution of PRSformer's non-linear Transformer architecture. It mirrors PRSformer precisely *except* for omitting the Transformer blocks:

   - **Identical data, inputs & training:** Uses the exact same $L \approx 140k$ input variants and individual-level train/validation/test splits. Employs the same multitask framework ($D = 18$), loss function, AdamW optimizer, and training schedule (Section 3.5). Uses the same embedding layer (Figure 1C) for genotypes and missingness.
   - **Architecture difference:** The input embeddings are fed *directly* to the final linear output layer, bypassing the Transformer blocks (Figure 1D).

   This provides a multitask linear model on the same large-scale individual data, allowing direct assessment of the performance gain from PRSformer's non-linear processing.

## 3.7 Cross-ancestry experiments

To assess generalization, we developed PRSformer-ME (Multi-Ethnic). We performed ancestry-specific GWAS (African American (AFR), European (EUR), Latino (LAT), East Asian (EAS), and South Asian (SAS); determined by internal classifier) using the same 08-2021 data freeze and variant selection criteria (Section 3.4, p-value $< 1 \times 10^{-2}$, QC) where sample sizes permitted (Supplementary Table F12). We defined an expanded input set ($L = 251,538$ variants) as the union of selected variants across all available disease-ancestry pairs (including Europeans). We constructed a multi-ancestry training set ($\sim$ 5M total individuals) by combining the European training set (Section 3.3) with $N = 1,136,746M$ non-European individuals meeting the same filtering criteria. PRSformer-ME was trained on this combined dataset using the same architecture and hyperparameters as the European-only PRSformer, without additional ancestry-specific tuning. Evaluation was performed on a combined test set including the European test set and held-out non-European individuals processed identically (Supplementary Table F3).

# 4 Experiments and results

We present results evaluating PRSformer's performance against baselines, analyzing the impact of non-linearity and sample scale, assessing the benefit of multitask learning, and testing cross-ancestry generalization using AUROC as the primary metric unless otherwise stated.

## 4.1 PRSformer outperforms state-of-the-art baselines

We first benchmarked PRSformer against the highly optimized linear model and the state-of-the-art summary-statistic method, LDPred2, on the European test set ($N_{test} \approx 494$k). As shown in Figure 2, PRSformer consistently achieves higher AUROC scores than its linear counterpart across all 18 autoimmune and inflammatory conditions. This comparison, using identical data and training setups except for the Transformer blocks, directly quantifies the predictive benefit derived from PRSformer's non-linear architecture.

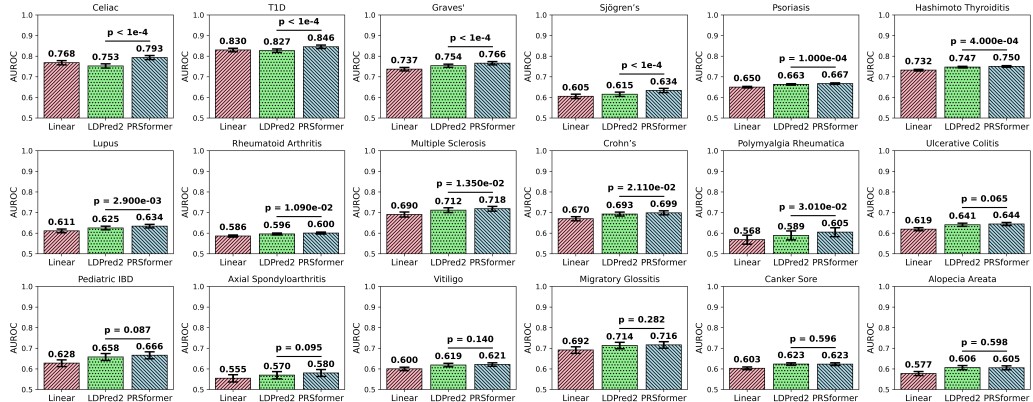

Figure 2: Benchmarking PRSformer against baseline methods using AUROC. PRSformer consistently outperforms the linear model (trained on identical individual-level data) and LDPred2 (state-of-the-art summary statistic method derived from the same cohort). Error bars: 95% CI (10k bootstraps). p-values: estimated using a one-sided paired bootstrap test (10,000 replicates), sampling with replacement from the test set and comparing AUROCs of PRSformer and LDPred2 on identical sample pairs. The p-value reflects the fraction of replicates where LDPred2's AUROC $\geq$ PRSformer's.

Crucially, PRSformer also significantly outperforms LDPred2 (using summary statistics derived from the *same* cohort) on 16 out of 18 diseases, with 11 differences being statistically significant ($p < 0.05$, one-sided paired bootstrap test). This demonstrates the advantage of end-to-end modeling on individual-level data compared to state-of-the-art methods relying on summary statistics. Consistent improvements were also observed in area under the precision–recall curve (Supplementary Figure E1) and explained variance (Supplementary Table F11), as well as when comparing against an enhanced Stacked LDPred2 baseline (PRSformer+, Supplementary Figure E2), confirming the robustness of PRSformer's advantage.

We also evaluated PRSformer and LDPred2 on the kinship-controlled test set, reproducing similar trends (Supplementary Figure E3): PRSformer outperformed LDPred2 in 14 of 18 diseases, maintaining its lead in 13 of the 16 and newly improving Alopecia Areata, with six remaining statistically significant ($p < 0.05$). The smaller number of significant improvements is expected given the reduced power of the kinship-controlled test set ($\sim$148k vs. $\sim$494k). These results confirm that PRSformer's advantage is not driven by familial confounding and persists under stringent kinship control, reinforcing the validity of our findings.

## 4.2 Benefit of non-linearity emerges at million-sample scale

To understand when the non-linear modeling capabilities of PRSformer become advantageous, we compared its performance against the linear baseline across varying training dataset sizes (down-sampling the $N \approx 3.8$M training set). Figure 3 reveals a critical insight: at smaller sample sizes, comparable to current large public cohorts like UK Biobank [53] (up to $N \approx 1$M), the performance

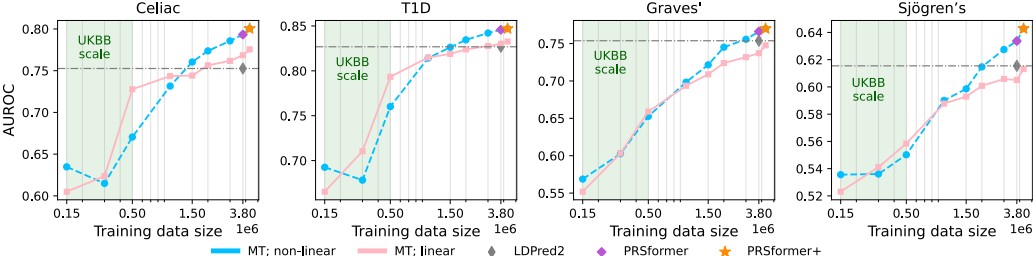

Figure 3: Impact of training scale on non-linear model advantage. Performance (AUROC) of PRSformer (non-linear) and the linear baseline across downsampled training sets (multitask setting). The benefit of non-linearity becomes apparent only at $N > 1M$ scale.

of PRSformer is similar to the simpler linear model. However, as the training size exceeds one million individuals, a clear advantage for the non-linear PRSformer emerges and progressively widens. This trend holds across multiple diseases (Supplementary Figure E4) and persists even when using appropriately subsetted variant sets for smaller scales (Supplementary Figure E5). These results indicate a scaling law: the benefits of non-linear architectures like PRSformer manifest only when sample sizes are sufficient to resolve higher-order genetic interactions. Below this threshold, linear models remain competitive, whereas beyond the million-sample regime, PRSformer achieves measurable gains (although with higher computational cost in FLOPs per sample).

## 4.3 Multitask learning consistently improves performance

We investigated the benefit of PRSformer's multitask design by comparing it against single-task (ST) models trained independently for each disease. Figure 4 shows that multitask (MT) training consistently yields superior AUROC compared to ST training for both PRSformer and the linear baseline, across different data scales (see Supplementary Figure E6 for other diseases). This improvement was robust even when ST models used disease-specific optimized variant sets (Supplementary Figure E7). Thus, the gain stems from leveraging shared information across related immune-mediated inflammatory diseases allowing shared model components (variant embeddings and Transformer blocks in PRSformer, and variant embedding in the linear baseline) to be optimized more effectively. By training these shared layers across multiple related diseases, the model can capture generalizable representations that enhance performance beyond what is achievable with isolated, ST training.

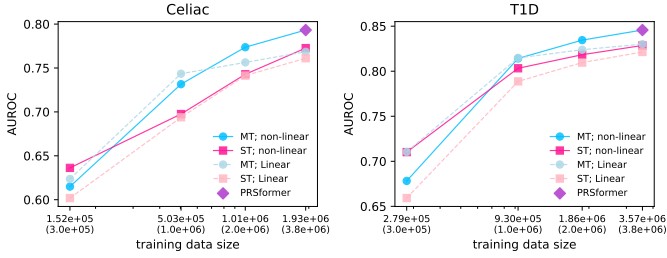

Figure 4: Multitask (MT) vs. Single-Task (ST) training for Celiac disease (left) and T1D (right). MT consistently outperforms ST for both non-linear (based on PRSformer) and the linear baseline across different scales (X-axis: ST sizes / (MT sizes)).

## 4.4 Improved cross-ancestry generalization via multitask multi-ancestry training

Recognizing the need for equitable genomic prediction [54], we trained PRSformer-ME on a combined multi-ancestry cohort ($\sim$ 5M individuals, including $\sim$ 1.1M non-Europeans) using an expanded variant set ($L \approx 252k$, see Section 3.7 Methods). We evaluated its performance on a held-out test set containing individuals from European (EUR), African American (AFR), Latino (LAT), East Asian (EAS), and South Asian (SAS) ancestries, comparing it to the original PRSformer trained only on EUR individuals.

As summarized in Table 1, PRSformer-ME demonstrates significantly improved generalization to non-EUR populations. It achieves substantially higher AUROC scores across most diseases in AFR, LAT, EAS, and SAS individuals compared to the EUR-only model. Importantly, this gain in non-EUR

populations is achieved with minimal to no degradation in performance on EUR individuals. These results indicate that training on diverse, aggregated individual-level data allows PRSformer-ME to capture both shared and ancestry-specific genetic signals, leading to more accurate and potentially more equitable predictions across populations compared to models trained on a single ancestry group (incorporating covariates such as sex and age further improves predictive performance, see Supplementary Table F14). This is an important finding since state-of-the-art methods generally rely on single-ancestry summary statistics, preventing them from jointly training on individual-level data across multiple ancestries and from leveraging shared cross-population genetic signals.

Table 1: AUROC of EUR-only PRSformer vs. multi-ancestry PRSformer-ME on the multi-ancestry test set. PRSformer-ME shows improved performance in non-EUR ancestries often without sacrificing EUR performance. Bold font denotes the higher AUROC in each pairwise comparison.

| Ances. | Model | Celiac | T1D | Graves' | Sjögren's | Psoriasis | Hashimoto Thyroiditis | Lupus | Rheumatoid Arthritis | Multiple Sclerosis | Crohn's | Polymyalgia Rheumatica | Ulcerative Colitis | Pediatric IBD | Axial Spondyl. | Vitiligo | Migratory Glossitis | Canker Sore | Alopecia Areata |
|---|---|---|---|---|---|---|---|---|---|---|---|---|---|---|---|---|---|---|---|
| EUR | PRSformer | **0.7933** | **0.8457** | 0.7661 | 0.6337 | 0.6669 | **0.7504** | 0.6338 | 0.6004 | **0.7184** | **0.6985** | **0.6048** | 0.6442 | **0.6660** | 0.5802 | 0.6212 | **0.7164** | 0.6227 | 0.6051 |
| | PRSformer-ME | 0.7867 | 0.8431 | **0.7674** | **0.6404** | **0.6683** | 0.7487 | **0.6458** | **0.6069** | 0.7157 | 0.6981 | 0.5995 | **0.6447** | 0.6654 | **0.5818** | **0.6222** | 0.7146 | **0.6246** | **0.6125** |
| AFR | PRSformer | 0.6219 | 0.6391 | 0.5676 | 0.5118 | 0.5559 | 0.6727 | 0.5404 | 0.5241 | 0.5447 | 0.5726 | – | 0.5401 | 0.5928 | <0.5 | 0.5122 | 0.6008 | 0.5913 | <0.5 |
| | PRSformer-ME | **0.6269** | **0.6986** | **0.6997** | **0.6399** | **0.5822** | **0.6939** | **0.6140** | **0.5660** | **0.6397** | **0.6217** | – | **0.5902** | **0.6285** | **0.5410** | **0.5605** | **0.6550** | **0.6026** | **0.5844** |
| LAT | PRSformer | 0.7437 | 0.7459 | 0.7060 | 0.6181 | 0.6449 | 0.7482 | 0.6063 | 0.5712 | 0.6832 | 0.6674 | 0.6526 | 0.6184 | 0.6216 | 0.5610 | 0.5871 | 0.6994 | 0.6283 | 0.5579 |
| | PRSformer-ME | **0.7521** | **0.7732** | **0.7568** | **0.6685** | **0.6596** | **0.7581** | **0.6536** | **0.5979** | **0.7256** | **0.6842** | **0.6553** | **0.6291** | **0.6539** | **0.6038** | **0.6166** | **0.7270** | **0.6397** | **0.6346** |
| SAS | PRSformer | **0.8180** | 0.6640 | **0.7667** | 0.6383 | 0.6209 | 0.6705 | 0.5656 | **0.6052** | 0.7341 | 0.6630 | – | **0.6974** | 0.6423 | **0.5999** | 0.5693 | **0.5421** | **0.5999** | 0.5754 |
| | PRSformer-ME | 0.8002 | **0.6852** | 0.7612 | **0.6885** | **0.6269** | **0.6907** | **0.6196** | 0.5974 | **0.7503** | **0.7061** | – | .6969 | **0.6858** | 0.5828 | **0.6009** | 0.5146 | 0.5813 | **0.5972** |
| EAS | PRSformer | | 0.6507 | 0.6937 | 0.5770 | 0.6167 | 0.6535 | 0.6144 | 0.5566 | **0.5406** | 0.6282 | **0.7431** | 0.6204 | 0.5902 | 0.5184 | 0.5463 | 0.6809 | 0.5731 | 0.5846 |
| | PRSformer-ME | – | **0.7121** | **0.7487** | **0.6176** | **0.6355** | **0.6876** | **0.6573** | **0.6085** | 0.5019 | **0.6256** | 0.6973 | **0.6545** | **0.6518** | **0.5860** | **0.5602** | **0.7085** | **0.5929** | **0.6386** |

## 5 Conclusion

We introduced PRSformer, a scalable Transformer architecture leveraging neighborhood attention to enable end-to-end, multitask disease prediction from population-scale individual genotypes ($N \approx 5\text{M}$, $L \approx 140\text{k}$). Our rigorous evaluation on a unique large private cohort, conducted under IRB and using consented research participant data, demonstrates that PRSformer significantly outperforms strong linear and state-of-the-art summary-statistic baselines (LDPred2) derived from the same cohort.

A key finding of this work is that the benefit of PRSformer's non-linear modeling for complex immune-mediated inflammatory diseases emerges primarily at the million-sample scale ($N > 1\text{M}$). This advantage varies across diseases, with traits like celiac disease and type 1 diabetes benefiting substantially from non-linear modeling to explain disease risk variance [55]. This scaling law, alongside our findings that multitask training improves performance and multi-ancestry data enhances generalization, establishes a new framework for genomic prediction.

While PRSformer advances predictive accuracy, its gains come with higher computational demands that may limit immediate clinical scalability. Furthermore, future work is required to develop interpretation methods to understand the learned non-linear interactions, which is essential for biological hypothesis generation and experimental validation. A key future direction is to extend the framework beyond a single disease domain to a phenome-scale setting spanning thousands of traits. This approach is motivated by widespread genetic pleiotropy, where a single variant can influence multiple, seemingly disparate conditions. A unified model could therefore capture the shared genetic underpinnings linking diverse biological systems, such as the contribution of immune pathways to neurodegeneration and cancer.

Our research prioritizes fairness across diverse populations and the responsible deployment of genomic models. Recognizing the sensitivity of genomic data, we have taken steps to balance transparency with participant privacy: we provide detailed methodological descriptions and have released our implementation code at https://github.com/23andMe/PRSformer; however, the data and trained models are not publicly available.

Taken together, our results and scaling analyses position PRSformer as a foundation for phenome-scale genetic risk modeling that can fully leverage genetic pleiotropy to improve prediction and generalization at population scale.

## Acknowledgments and Disclosure of Funding

The authors thank the past and present employees and research participants of 23andMe for making this work possible. We are grateful to Akele Reed, Teague Sterling, David Hinds, Steve Pitts, Wei Wang, Bertram Koelsch, Michael Holmes, Stella Aslibekyan, Cordell Blakkan and Barry Hicks for their valuable contributions and insightful comments on the manuscript, and to Ali Hassani for helpful discussions on employing Neighborhood Attention. The authors also gratefully acknowledge the support of AWS for providing GPU computing resources and credits. A. A. Khan is supported in part by a Chan Zuckerberg Investigator Award.

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

## A    Details of the GWAS runs

The internal ancestry classifier assigns individuals to one of five major genetic ancestry groups - African American (AFR), European (EUR), East Asian (EAS), South Asian (SAS), or Latino (LAT) - based on local ancestry inference [48]. To reduce confounding introduced by population structure, GWAS analyses were stratified by these genetically inferred ancestry groups.

Principal component analysis (PCA) was conducted separately within each ancestry group using a subset of <100,000 high-quality genotyped variants shared across all internal platforms. A randomly selected subset of individuals was used for each group: 513K for AFR, 398K for EAS, 1M for EUR, 1M for LAT, and 111K for SAS [48].

For each disease and ancestry group, GWAS was performed using logistic regression with additive allelic effects as predictors. Covariates included age, sex, genotype platform (to adjust for batch effects), and top principal components - specifically, the top 5 PCs for EUR, EAS, and SAS; the top 6 for AFR; and the top 9 for LAT. Association p-values were derived using a likelihood ratio test, comparing a reduced model fitted using covariates only to a full model fitted with both additive genetic effects and covariates [56].

## B    Exploration of LD-based variant pruning

We additionally experimented with training PRSformer on a subset of variants that had been LD-pruned using *PLINK 2.0* [57]. LD pruning removes highly correlated SNPs to retain approximately independent markers. In this procedure, a sliding window is moved across the genome, pairwise linkage disequilibrium ($r^2$) is computed among variants, and SNPs exceeding a specified correlation threshold with nearby variants are iteratively removed until no pair within each window remains above that threshold. Starting from a union variant set constructed similarly to that in Section 3.4 (but with slightly adjusted filtering thresholds), we applied *PLINK 2.0* with a window size of 6,000 kb (6 Mb), a step size of one variant, and an $r^2$ threshold of 0.5. Supplementary Table F11 compares two models from the hyperparameter tuning round trained with and without LD-based variant pruning. Interestingly, despite reducing multicollinearity among variant features, LD pruning led to lower validation performance, suggesting that PRSformer benefits from leveraging the local correlation structure within LD blocks to capture causal signals more effectively.

## C    Details of LDpred2 runs

For each disease we used the Gibbs sampler LDpred2 software [5] on the summary statistics with an internal LD panel. LD matrix computation included variants with minor allele frequency greater than $0.1\%$, and genotype call rate greater than $90\%$. Variants greater than 5cM apart were assumed to be independent. Summary statistics were filtered to keep variants that had a minor allele frequency greater than $0.1\%$ and had a genotype call rate greater than $95\%$. This consisted of variant sets with roughly 445,000 variants. We estimated posterior SNP-effect sizes using the grid option with a set of 100 combinations of hyperparameters, leading to up to 100 sets of polygenic risk scores (PRS) per disease (depending on convergence). The hyperparameters that LDpred2 takes are an estimate for the proportion of causal variants, $p$, and trait heritability, $h^2$. We used LD score regression to estimate $h^2$, then used a grid of the $h^2$ estimate multiplied by 0.6, 0.8, 1, 1.2, and 1.4. For $p$, we used a sequence of values equally spaced on a logarithmic scale from $10^{-5}$ to 1. The best hyperparameters for each disease were selected based on validation AUROC leading to the final LDPred2 PRS models. For Stacked LDPred2, however, we ensembled all of the converged PRSs per disease (up to 100) by training elastic net on the validation data using 5-fold cross validation.

## D    Subsetting variants for down-sampled experiments

When training on smaller datasets, we may not have access to the same high-powered variant selection as in the full-data setting. To account for this, we repeated variant selection using GWAS summary statistics adjusted to reflect the reduced sample size of each downsampled dataset. For each downsampled dataset, we estimated GWAS p-values under the reduced sample size using the original

GWAS summary statistics:

$$Z = \frac{\beta}{SE}; \quad Z_{ds} = Z \times \sqrt{\frac{N_{ds}}{N}}; \quad p_{ds} = 2\Phi(-|Z_{ds}|)$$

where $\beta$ and $SE$ denote the effect size and standard error from the original GWAS, $N$ is the original sample size, $N_{ds}$ is the downsampled sample size, and $\Phi$ is the standard normal cumulative distribution function. Subsequently, variant selection was performed independently for each disease and each downsampled dataset using the estimated p-values, applying a threshold of $p < 1e-2$. Multi-task model training was then conducted using the union of the selected variant sets across diseases at each downsampled scale, following the same procedure as in the full-data experiments. Additional details on variant sets and data sizes and model configurations are provided in Supplementary Tables F1,F2 and F13.

# E   Supplementary figures

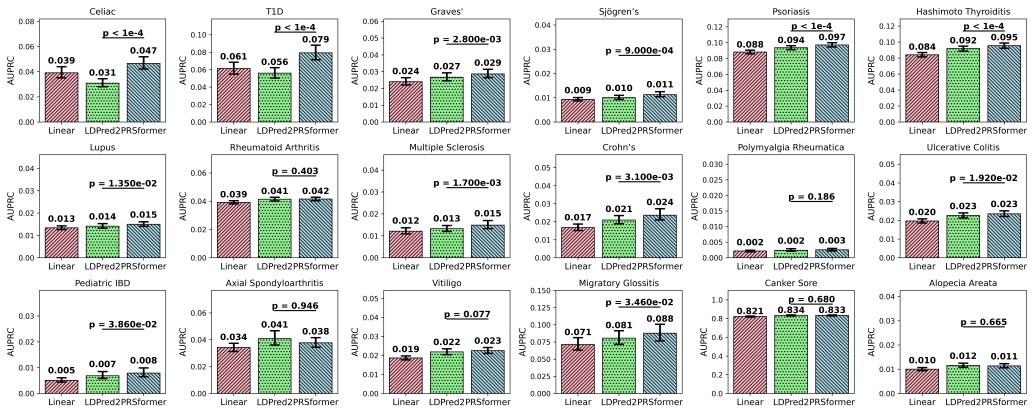

Figure E1: Benchmarking PRSformer against baseline methods using AUPRC as the evaluation metric. Numbers above the bars indicate test set AUPRC values; error bars denote 95% confidence intervals estimated via bootstrapped test samples. The reported p-values reflect the one-sided statistical significance of PRSformer outperforming LDPred2.

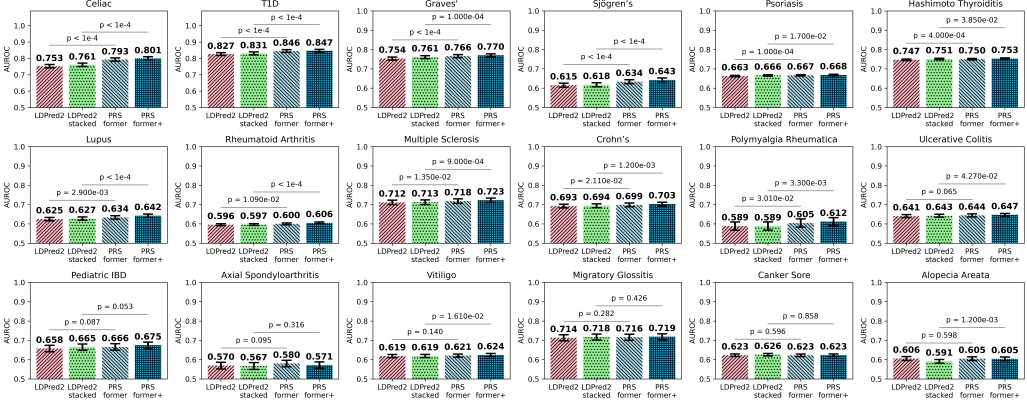

Figure E2: Benchmarking of models trained on the combined training and validation datasets. Numbers above the bars indicate test set AUROC; error bars represent 95% confidence intervals computed via bootstrapped test samples. The two sets of p-values reflect the one-sided statistical significance of PRSformer+ outperforming stacked LDPred2, and PRSformer outperforming non-stacked LDPred2—the latter being the same as those reported in Figure 2 of the main text.

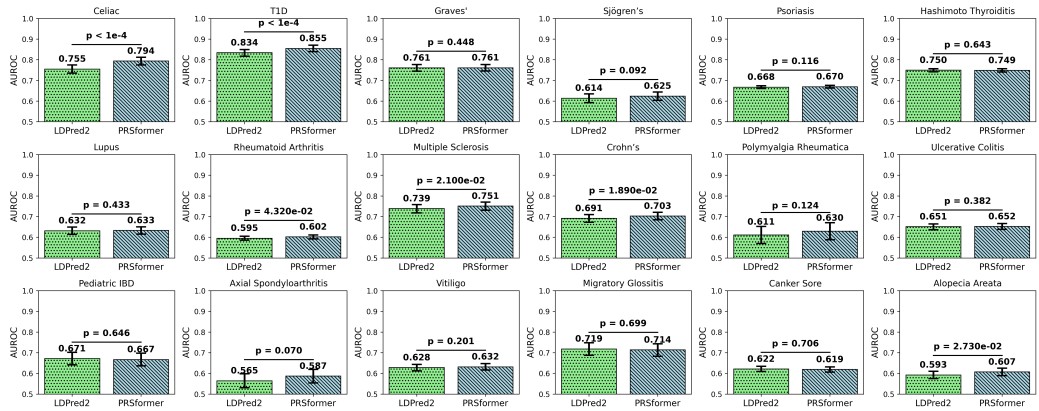

Figure E3: Comparison of PRSformer and LDPred2 on the kinship-controlled European test set, evaluated by AUROC. Numbers above the bars indicate AUROC values, and error bars represent 95% confidence intervals estimated from 10,000 bootstrapped samples. The reported p-values reflect the one-sided statistical significance of PRSformer outperforming LDPred2.

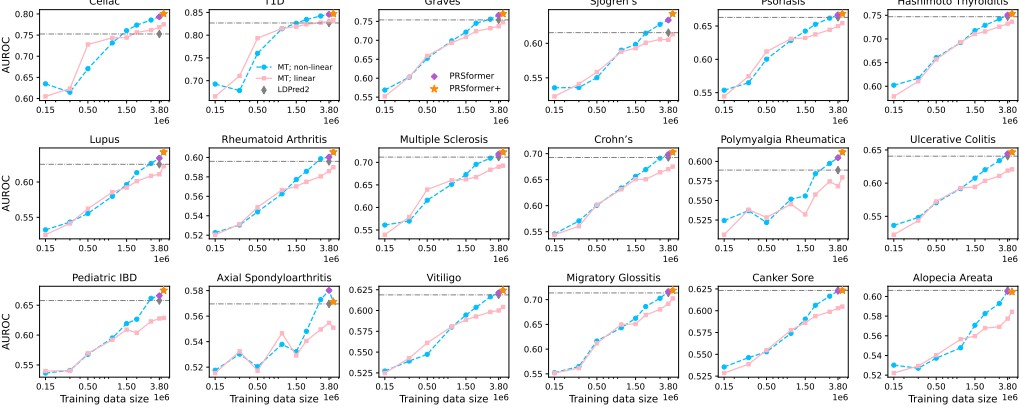

Figure E4: Prediction performance across different training scales for the non-linear model and linear baseline, both trained in a multitask (MT) setting using the same input variant set as PRSFormer. For most diseases, performance improves with more training data, with the non-linear model surpassing the linear baseline at larger scales. Fluctuations in performance for Polymyalgia Rheumatica and Axial Spondyloarthritis likely stem from the former's rarity and the latter's relatively small training size (see Supplementary Table F2).

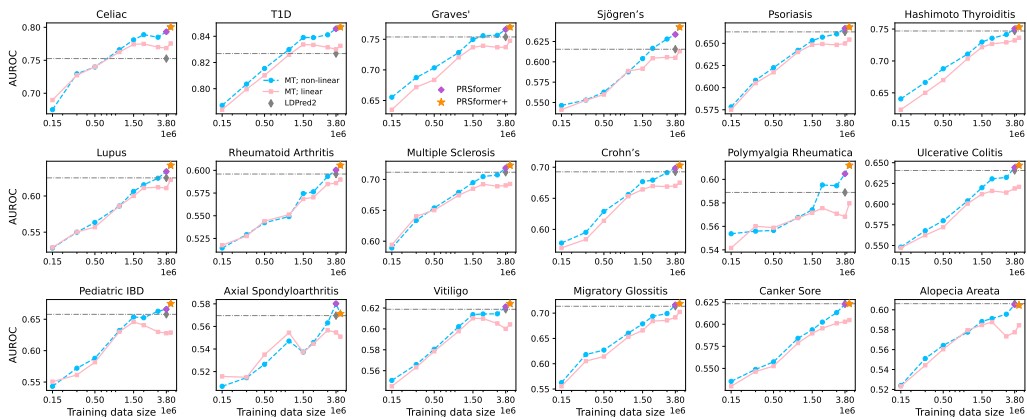

Figure E5: Prediction performance across different training scales for the non-linear model and linear baseline, both trained in a multitask (MT) setting using subsetted variant sets at each down-sampled scale (see Supplementary section D and Table F13). For most diseases, performance improves with increasing training data, with the non-linear model outperforming the linear baseline at larger scales. These trends are consistent with those observed using a fixed input variant set across scales (see Supplementary Figure E4).

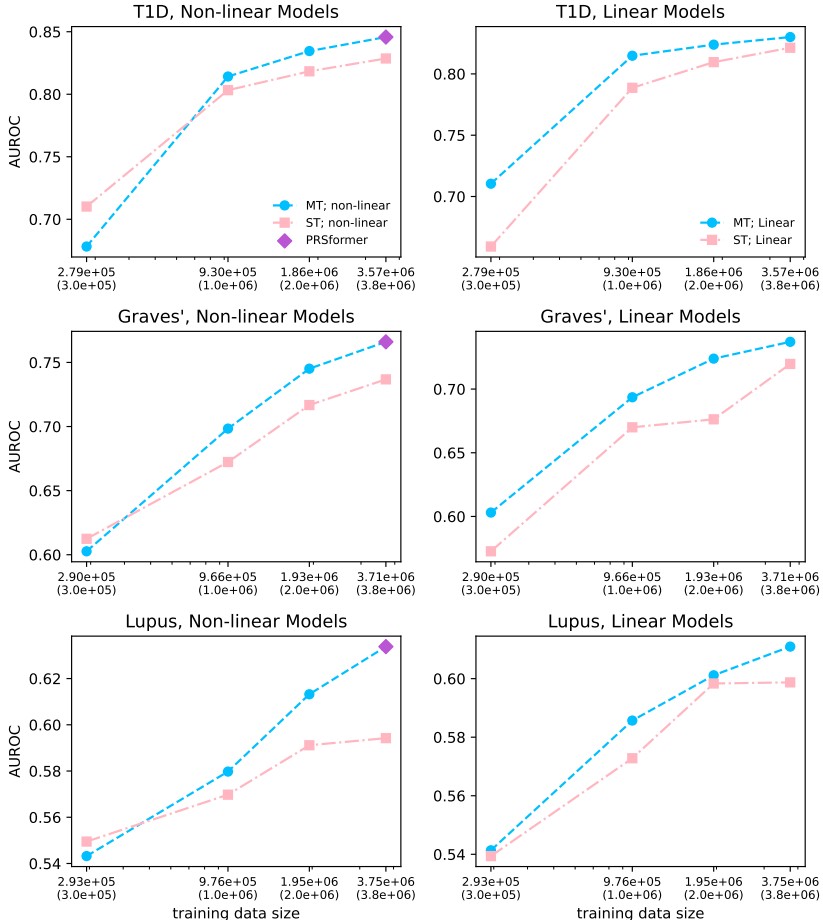

Figure E6: Comparison of multitask (MT) versus single-task (ST) training for three additional diseases across different training scales, all using the same input variant set as PRSformer. X-axis values outside parentheses indicate ST training sizes, and those inside indicate corresponding MT training sizes. Across all tested diseases, MT training outperforms ST training for both the non-linear model (left) and the linear baseline (right).

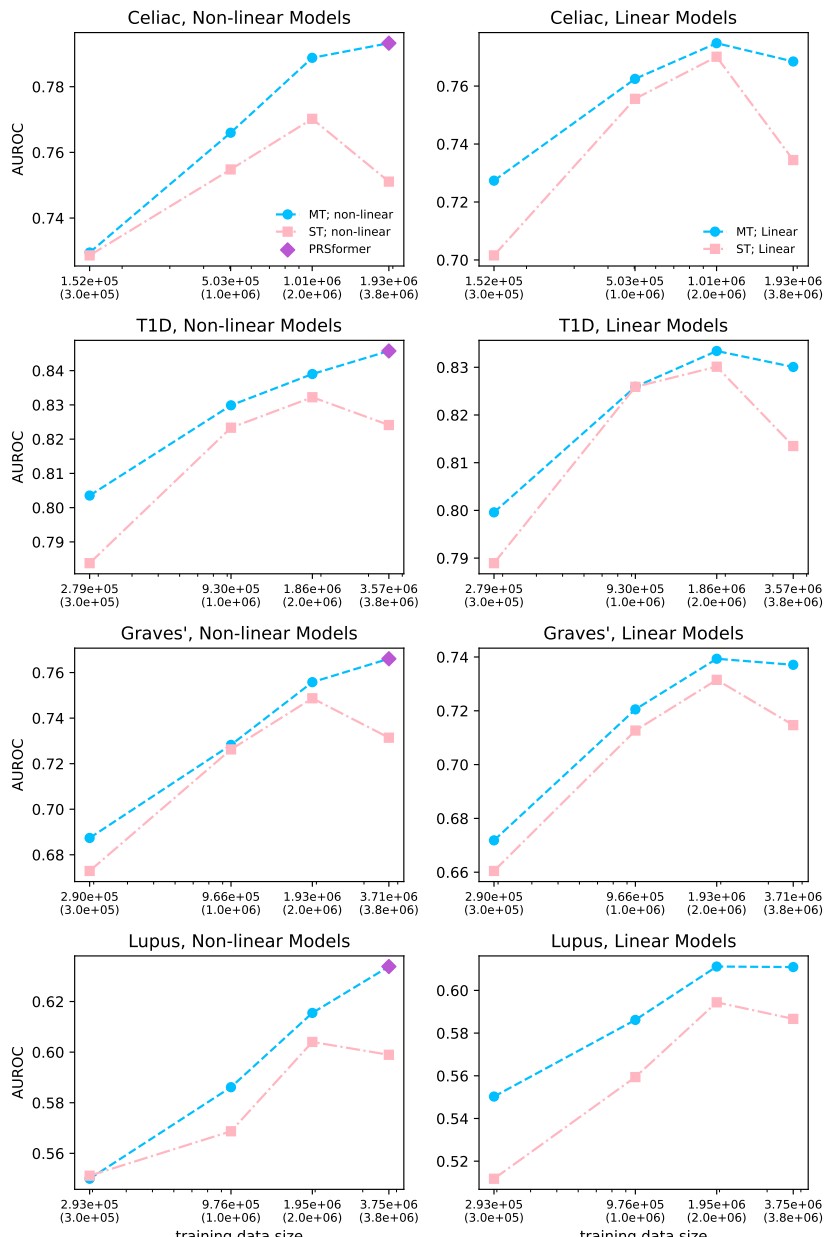

Figure E7: Comparison of multitask (MT) versus single-task (ST) training for four tested diseases across different training scales. Both MT and ST models were trained on downsampled datasets using subsetted variants; additionally, ST models used disease-specific variant sets (see Supplementary section D and Table F13). X-axis values outside parentheses indicate ST training sizes, while those inside indicate the corresponding MT sizes. Across all tested diseases, MT training outperforms ST training for both the non-linear model (left) and the linear baseline (right). These results are consistent with those observed using a fixed input variant set for both MT and ST across scales (see Supplementary Figure E6).

# F   Supplementary tables

Table F1: Number of selected variants per disease across European datasets

| Disease | EUR dataset | 3M | 2M | 1.5M | 1M | 0.5M | 0.3M | 0.15M |
|---|---|---|---|---|---|---|---|---|
| Celiac | 14291 | 9595 | 6461 | 5576 | 4923 | 3906 | 3380 | 2722 |
| T1D | 16176 | 12607 | 7865 | 6424 | 5323 | 4033 | 3371 | 2530 |
| Graves' | 17350 | 14104 | 8380 | 6614 | 5231 | 3725 | 2984 | 2096 |
| Sjögren's | 11949 | 9381 | 4692 | 3502 | 2633 | 1954 | 1439 | 845 |
| Psoriasis | 24685 | 19621 | 11943 | 9088 | 6770 | 4529 | 3478 | 2148 |
| Hashimoto Thyroiditis | 29806 | 25103 | 15141 | 11314 | 7869 | 4677 | 3256 | 2030 |
| Lupus | 13618 | 10252 | 4840 | 3305 | 2384 | 1512 | 943 | 494 |
| Rheumatoid Arthritis | 18426 | 13683 | 6303 | 3980 | 2497 | 1484 | 891 | 373 |
| Multiple Sclerosis | 13347 | 10124 | 5587 | 4309 | 3301 | 2328 | 1683 | 1226 |
| Crohn's | 14140 | 10444 | 5126 | 3397 | 2099 | 739 | 302 | 90 |
| Polymyalgia Rheumatica | 8113 | 5818 | 2440 | 1674 | 1135 | 558 | 267 | 78 |
| Ulcerative Colitis | 14138 | 10381 | 5081 | 3487 | 2300 | 1042 | 547 | 164 |
| Pediatric IBD | 8677 | 5679 | 1954 | 1095 | 650 | 204 | 72 | 24 |
| Axial Spondyloarthritis | 6352 | 2215 | 668 | 316 | 122 | 20 | 4 | 0 |
| Vitiligo | 12980 | 9720 | 5231 | 3926 | 2900 | 1695 | 1092 | 525 |
| Migratory Glossitis | 14309 | 8692 | 4516 | 3327 | 2394 | 1457 | 1019 | 610 |
| Canker Sore | 18418 | 6181 | 3204 | 2126 | 1322 | 485 | 206 | 34 |
| Alopecia Areata | 9741 | 7117 | 3160 | 2087 | 1325 | 680 | 312 | 59 |
| Union | 137245 | 94176 | 38018 | 23397 | 15060 | 9181 | 6829 | 4863 |

Table F2: Training sample sizes (case / control) per disease across European and multi-ancestry datasets

| Disease | Multi-Ancestry Dataset | Full Dataset | 3M | 2M | 1.5M | 1M | 0.5M | 0.3M | 0.15M |
|---|---|---|---|---|---|---|---|---|---|
| Celiac | 14851 / 2444838 | 13304 / 1933687 | 10451 / 1511685 | 6828 / 1007277 | 5228 / 755823 | 3458 / 503346 | 1747 / 251838 | 1052 / 151689 | 555 / 75772 |
| T1D | 25815 / 4600428 | 21108 / 3568017 | 16519 / 2788978 | 11126 / 1859420 | 8246 / 1393969 | 5525 / 929582 | 2714 / 464434 | 1626 / 278712 | 808 / 139611 |
| Graves' | 34377 / 4767527 | 26982 / 3708583 | 21132 / 2898733 | 14099 / 1932335 | 10573 / 1449024 | 7154 / 966394 | 3518 / 482997 | 2131 / 289748 | 1043 / 144919 |
| Sjögren's | 23480 / 4791439 | 19544 / 3718113 | 15259 / 2906128 | 10245 / 1937041 | 7562 / 1452756 | 5032 / 968626 | 2588 / 484220 | 1497 / 290448 | 735 / 145343 |
| Psoriasis | 181096 / 4404301 | 151767 / 3560128 | 118647 / 2782686 | 79073 / 1854947 | 59461 / 1391012 | 39630 / 927636 | 19622 / 463485 | 11922 / 278024 | 5934 / 139104 |
| Hashimoto Thyroiditis | 124104 / 4677800 | 106911 / 3708583 | 83363 / 2898733 | 55523 / 1932335 | 41881 / 1449024 | 27653 / 966394 | 14022 / 482997 | 8314 / 289748 | 4172 / 144919 |
| Lupus | 35926 / 4816513 | 27795 / 3747933 | 21623 / 2929376 | 14499 / 1952708 | 10807 / 1464450 | 7293 / 976354 | 3661 / 488102 | 2144 / 292815 | 1092 / 146446 |
| Rheumatoid Arthritis | 128990 / 4532187 | 104481 / 3611921 | 81737 / 2823204 | 54452 / 1882141 | 40607 / 1411177 | 27066 / 941019 | 13610 / 470158 | 8174 / 282097 | 3970 / 141143 |
| Multiple Sclerosis | 20230 / 4812927 | 17015 / 3735898 | 13384 / 2920007 | 8962 / 1946527 | 6689 / 1459824 | 4390 / 973437 | 2170 / 486450 | 1329 / 291870 | 639 / 145979 |
| Crohn's | 27434 / 4667812 | 23802 / 3637333 | 18571 / 2842879 | 12475 / 1895460 | 9301 / 1421090 | 5975 / 947547 | 3210 / 473495 | 1952 / 284047 | 992 / 142142 |
| Polymyalgia Rheumatica | 7469 / 4620931 | 6940 / 3588209 | 5392 / 2804710 | 3597 / 1869745 | 2742 / 1401959 | 1802 / 934872 | 880 / 467030 | 557 / 280181 | 274 / 140198 |
| Ulcerative Colitis | 53021 / 4636460 | 44568 / 3632871 | 34799 / 2839378 | 23012 / 1893078 | 17467 / 1419395 | 11663 / 946417 | 5880 / 472898 | 3456 / 283722 | 1784 / 141974 |
| Pediatric IBD | 10014 / 4613960 | 8311 / 3577312 | 6470 / 2796009 | 4303 / 1864258 | 3316 / 1397699 | 2145 / 932056 | 1143 / 465543 | 639 / 279336 | 337 / 139724 |
| Axial Spondyloarthritis | 7786 / 336188 | 6843 / 292915 | 5377 / 229150 | 3527 / 153102 | 2682 / 114299 | 1812 / 76600 | 901 / 37712 | 532 / 22931 | 258 / 11477 |
| Vitiligo | 38209 / 4534376 | 29309 / 3550146 | 22855 / 2774956 | 15261 / 1849633 | 11479 / 1387067 | 7725 / 925064 | 3682 / 462197 | 2295 / 277303 | 1159 / 138722 |
| Migratory Glossitis | 32394 / 1217541 | 27982 / 1031326 | 21764 / 805875 | 14687 / 537379 | 10837 / 402928 | 7404 / 268314 | 3627 / 134668 | 2176 / 80804 | 1122 / 40505 |
| Canker Sore | 578838 / 194610 | 493951 / 643681 | 385900 / 502693 | 257526 / 335273 | 193006 / 251409 | 128495 / 167596 | 64644 / 84032 | 38620 / 50224 | 19452 / 25453 |
| Alopecia Areata | 35533 / 4498787 | 22506 / 3522691 | 17705 / 2753494 | 11820 / 1835462 | 8740 / 1376318 | 5895 / 917773 | 2885 / 458592 | 1725 / 275141 | 809 / 137684 |

Table F3: Test data sample sizes (case / control) per disease across ancestry groups

| Disease | EUR | AFR | LAT | EAS | SAS |
|---|---|---|---|---|---|
| Celiac | 2160 / 243495 | 80 / 46973 | 630 / 172120 | – | 30 / 8072 |
| T1D | 2720 / 457259 | 615 / 105176 | 1391 / 309332 | 77 / 55295 | 48 / 17309 |
| Graves' | 3388 / 469283 | 914 / 110197 | 1766 / 317074 | 422 / 55642 | 45 / 17946 |
| Sjögren's | 2816 / 472870 | 520 / 111322 | 1310 / 319124 | 157 / 56119 | 50 / 18187 |
| Psoriasis | 21015 / 431120 | 2521 / 101166 | 9615 / 292205 | 1530 / 50308 | 569 / 16048 |
| Hashimoto Thyroiditis | 14186 / 458485 | 804 / 110307 | 6093 / 312747 | 367 / 55697 | 317 / 17674 |
| Lupus | 3936 / 473772 | 1137 / 111165 | 2442 / 319080 | 225 / 56313 | 72 / 18237 |
| Rheumatoid Arthritis | 13184 / 443140 | 3280 / 101623 | 7071 / 297862 | 507 / 52176 | 188 / 16702 |
| Multiple Sclerosis | 2025 / 446886 | 475 / 101646 | 780 / 297888 | 36 / 51393 | 29 / 16457 |
| Crohn's | 3005 / 455308 | 513 / 105068 | 1033 / 305040 | 67 / 53528 | 66 / 16929 |
| Polymyalgia Rheumatica | 708 / 435628 | – | 124 / 287923 | 10 / 49222 | – |
| Ulcerative Colitis | 5298 / 452158 | 796 / 104667 | 2518 / 303129 | 192 / 52713 | 147 / 16845 |
| Pediatric IBD | 1147 / 449546 | 191 / 104216 | 546 / 302110 | 33 / 52611 | 37 / 16782 |
| Axial Spondyloarthritis | 1086 / 36301 | 47 / 3144 | 363 / 16630 | 53 / 2073 | 22 / 503 |
| Vitiligo | 5109 / 446859 | 1246 / 102158 | 3199 / 298421 | 298 / 51520 | 214 / 16422 |
| Migratory Glossitis | 1092 / 36473 | 108 / 4993 | 358 / 18007 | 39 / 2133 | 11 / 565 |
| Canker Sore | 28275 / 8842 | 2442 / 2463 | 11662 / 6178 | 1579 / 596 | 284 / 272 |
| Alopecia Areata | 3263 / 445400 | 2198 / 100221 | 3527 / 294859 | 606 / 50802 | 358 / 15913 |

Table F4: Validation dataset sample sizes (case/control) per disease (only European individuals)

| Disease | EUR Validation Size |
|---|---|
| Celiac | 2096 / 262416 |
| T1D | 3088 / 488880 |
| Graves' | 3469 / 506279 |
| Sjögren's | 2964 / 509623 |
| Psoriasis | 22883 / 467419 |
| Hashimoto Thyroiditis | 14555 / 495193 |
| Lupus | 4173 / 511037 |
| Rheumatoid Arthritis | 14315 / 479237 |
| Multiple Sclerosis | 2203 / 484295 |
| Crohn's | 3449 / 493246 |
| Polymyalgia Rheumatica | 662 / 472910 |
| Ulcerative Colitis | 5966 / 489957 |
| Pediatric IBD | 1285 / 487101 |
| Axial Spondyloarthritis | 1153 / 39350 |
| Vitiligo | 5320 / 484047 |
| Migratory Glossitis | 1374 / 46592 |
| Canker Sore | 36835 / 11355 |
| Alopecia Areata | 3567 / 482771 |

Table F5: Tuning of the training steps based on validation AUROC

| Model | Epoch | Celiac | Crohn's | Graves' | Hashimoto Thyroiditis | Lupus | Multiple Sclerosis | Psoriasis | Rheumatoid Arthritis | Sjögren's | T1D |
|---|---|---|---|---|---|---|---|---|---|---|---|
| model-t059 | 2.0 | **0.7732** | **0.6843** | **0.7621** | **0.7412** | **0.6295** | **0.7165** | **0.6593** | **0.5911** | **0.6401** | **0.8392** |
| model-t059 | 3.0 | 0.7368 | 0.6592 | 0.7323 | 0.7225 | 0.6087 | 0.6829 | 0.6415 | 0.5780 | 0.6164 | 0.8162 |
| model-t15 | 1.8 | 0.7775 | 0.6869 | 0.7623 | 0.7417 | 0.6279 | 0.7137 | 0.6581 | 0.5901 | 0.6357 | 0.8398 |
| model-t16 | 1.9 | **0.7785** | 0.6895 | 0.7632 | 0.7435 | **0.6345** | 0.7173 | 0.6600 | 0.5924 | 0.6371 | 0.8404 |
| model-t17 | 2.0 | 0.7784 | **0.6904** | **0.7648** | **0.7446** | 0.6341 | **0.7218** | **0.6613** | **0.5939** | 0.6405 | **0.8424** |
| model-t18 | 2.1 | 0.7463 | 0.6717 | 0.7386 | 0.7235 | 0.6147 | 0.6905 | 0.6410 | 0.5782 | 0.6164 | 0.8173 |
| model-t19 | 2.2 | 0.7430 | 0.6666 | 0.7341 | 0.7193 | 0.6099 | 0.6829 | 0.6376 | 0.5758 | 0.6132 | 0.8132 |

*Note.* Most models (e.g., model-t059) showed signs of overfitting beyond 2 training epochs. We also explored training durations around this point (1.8–2.2 epochs) and selected 2 epochs as the final configuration.

Table F6: Tuning of attention heads and model dimension based on validation AUROC

| Model | #Atten. Head | #d_model | Celiac | Crohn's | Graves' | Hashimoto Thyroiditis | Lupus | Multiple Sclerosis | Psoriasis | Rheumatoid Arthritis | Sjögren's | T1D | Mean |
|---|---|---|---|---|---|---|---|---|---|---|---|---|---|
| Model-t17 | 4 | 64 | **0.7784** | 0.6904 | 0.7648 | 0.7446 | 0.6341 | **0.7218** | 0.6613 | 0.5939 | 0.6405 | **0.8424** | **0.7072** |
| Model-t20 | 4 | 32 | 0.7649 | 0.6794 | 0.7568 | 0.7418 | 0.6214 | 0.7081 | 0.6584 | 0.5902 | 0.6343 | 0.8371 | 0.6992 |
| Model-t21 | 4 | 96 | 0.7754 | 0.6892 | **0.7656** | **0.7464** | 0.6334 | 0.7208 | **0.6628** | **0.5954** | **0.6427** | 0.8424 | **0.7074** |
| Model-t22 | 3 | 48 | 0.7783 | **0.6909** | 0.7640 | 0.7449 | **0.6357** | 0.7192 | 0.6612 | 0.5927 | 0.6408 | 0.8424 | 0.7070 |
| Model-t23 | 3 | 24 | 0.7714 | 0.6819 | 0.7527 | 0.7404 | 0.6259 | 0.7136 | 0.6558 | 0.5897 | 0.6315 | 0.8392 | 0.7002 |
| Model-t24 | 3 | 72 | 0.7667 | 0.6865 | 0.7649 | 0.7448 | 0.6289 | 0.7131 | 0.6611 | 0.5932 | 0.6403 | 0.8397 | 0.7039 |
| Model-t25 | 5 | 80 | 0.7659 | 0.6871 | 0.7643 | 0.7446 | 0.6278 | 0.7127 | 0.6613 | 0.5941 | 0.6406 | 0.8405 | 0.7039 |
| Model-t26 | 5 | 40 | 0.7741 | 0.6874 | 0.7602 | 0.7426 | 0.6309 | 0.7145 | 0.6591 | 0.5912 | 0.6377 | 0.8401 | 0.7038 |

*Note.* We selected 4 attention heads with $d_{\mathrm{model}} = 64$ for their competitive performance despite a smaller model dimension compared to Model-t21.

Table F7: Tuning of attention dilation and number of transformer blocks based on validation AUROC

| Model | #Transformer Blocks | NA Dilation | Celiac | Crohn's | Graves' | Hashimoto Thyroiditis | Lupus | Multiple Sclerosis | Psoriasis | Rheumatoid Arthritis | Sjögren's | T1D | Mean |
|---|---|---|---|---|---|---|---|---|---|---|---|---|---|
| Model-t9 | 2 | (1-1) | 0.7807 | **0.6938** | **0.7772** | **0.7508** | **0.6422** | **0.7297** | 0.6654 | 0.5987 | **0.6501** | 0.8425 | 0.7131 |
| Model-t12 | 2 | (1-2) | 0.7806 | 0.6904 | 0.7766 | 0.7507 | 0.6401 | 0.7292 | **0.6659** | **0.5991** | 0.6497 | **0.8435** | 0.7126 |
| Model-t13 | 3 | (1-2-3) | **0.7808** | 0.6921 | 0.7749 | 0.7498 | 0.6398 | 0.7291 | 0.6650 | 0.5982 | 0.6459 | 0.8427 | 0.7118 |

*Note.* Increasing the number of transformer blocks and applying dilated attention (e.g., (1–2), (1–2–3)) led to mild overfitting. We selected 2 transformer blocks without dilation (1–1) as the final configuration.

Table F8: Tuning of the window size of neighborhood attention based on validation AUROC

| Model | #Atten. Head | #NA Window | Celiac | Crohn's | Graves' | Hashimoto Thyroiditis | Lupus | Multiple Sclerosis | Psoriasis | Rheumatoid Arthritis | Sjögren's | T1D |
|---|---|---|---|---|---|---|---|---|---|---|---|---|
| Model-t75 | 4 | 385 | 0.7806 | 0.6944 | **0.7700** | **0.7415** | 0.6308 | **0.7265** | **0.6625** | **0.5975** | 0.6311 | **0.8418** |
| Model-t127 | 4 | 129 | 0.7796 | **0.6955** | 0.7699 | 0.7410 | **0.6310** | 0.7265 | 0.6618 | 0.5963 | 0.6322 | 0.8412 |
| Model-t128 | 4 | 257 | **0.7821** | 0.6922 | 0.7679 | 0.7403 | 0.6299 | 0.7246 | 0.6610 | 0.5963 | 0.6303 | 0.8410 |
| Model-t129 | 4 | 513 | 0.7798 | 0.6945 | 0.7684 | 0.7412 | 0.6292 | 0.7252 | 0.6619 | 0.5969 | 0.6305 | 0.8413 |
| Model-t130 | 4 | 641 | 0.7787 | 0.6940 | 0.7691 | 0.7411 | 0.6308 | 0.7233 | **0.6625** | 0.5959 | **0.6342** | 0.8405 |
| Model-t131 | 2 | 129 | 0.7799 | 0.6927 | 0.7670 | 0.7398 | 0.6298 | 0.7252 | 0.6615 | 0.5954 | 0.6297 | 0.8403 |
| Model-t132 | 2 | 257 | 0.7785 | 0.6936 | 0.7676 | 0.7408 | 0.6303 | 0.7234 | 0.6617 | 0.5962 | 0.6315 | 0.8404 |
| Model-t133 | 2 | 513 | 0.7799 | 0.6946 | 0.7694 | 0.7413 | 0.6297 | 0.7256 | 0.6619 | 0.5961 | 0.6319 | 0.8410 |
| Model-t134 | 2 | 641 | 0.7807 | 0.6923 | 0.7668 | 0.7404 | 0.6286 | 0.7247 | 0.6618 | 0.5948 | 0.6307 | 0.8407 |

*Note.* We selected 4 attention heads with Neighborhood Attention's window size of 385 as the final configuration.

Table F9: Exploring the impact of LD-based variant set pruning ($r^2 = 0.5$) on validation AUROC.

| Model | No. variants | LD-pruned | Celiac | Crohn's | Graves' | Hashimoto Thyroiditis | Lupus | Multiple Sclerosis | Psoriasis | Rheumatoid Arthritis | Sjögren's | T1D |
|---|---|---|---|---|---|---|---|---|---|---|---|---|
| model-t031 | ~344K | No | **0.7640** | **0.6103** | **0.7334** | **0.7335** | **0.6192** | **0.6717** | **0.6572** | **0.5755** | **0.6340** | **0.8328** |
| model-t034 | ~232K | Yes | 0.7564 | 0.5951 | 0.7154 | 0.7179 | 0.6143 | 0.6595 | 0.6466 | 0.5732 | 0.6300 | 0.8238 |

*Note.* Interestingly, removing correlated variants via LD pruning lowers AUROCs, indicating that PRSformer benefits from the underlying LD structure when identifying causal signals.

Table F10: Exploring different output heads based on validation AUROC.

| Model | Output Head | Celiac | Crohn's | Graves' | Hashimoto Thyroiditis | Lupus | Multiple Sclerosis | Psoriasis | Rheumatoid Arthritis | Sjögren's | T1D |
|---|---|---|---|---|---|---|---|---|---|---|---|
| model-t5 | Flatten+FC | **0.7799** | **0.6896** | **0.7773** | **0.7497** | **0.6385** | **0.7301** | **0.6655** | **0.5981** | **0.6491** | **0.8448** |
| model-t10 | 10-[CLS]-tokens+Flatten+FC | 0.7343 | 0.5678 | 0.6656 | 0.6682 | 0.5840 | 0.6410 | 0.6036 | 0.5590 | 0.6007 | 0.8004 |
| model-t11 | Global-Avg-Pool+FC | 0.5820 | 0.5348 | 0.5955 | 0.5822 | 0.5516 | 0.5187 | 0.5504 | 0.5292 | 0.5601 | 0.5996 |

*Note.* In model-t10, we introduced ten [CLS] tokens into the input sequence and vocabulary, each with learnable embeddings and full attention over all variant tokens. The flattened, normalized representations of these [CLS] tokens were passed to a linear layer producing an 18-dimensional output. In model-t11, an average-pooling layer was applied to the normalized representations from the last transformer block (reducing tensors from $B \times L \times d$ to $B \times d$), followed by a linear layer mapping $d$ to the 18-dimensional output.

Table F11: Comparison of explained variance between PRSformer and LDpred2 across diseases on the European test set

| Model | Celiac | T1D | Graves' | Sjögren's | Psoriasis | Hashimoto Thyroiditis | Lupus | Rheumatoid Arthritis | Multiple Sclerosis | Crohn's | Polymyalgia Rheumatica | Ulcerative Colitis | Pediatric IBD | Axial Spondyloarthritis | Vitiligo | Migratory Glossitis | Canker Sore | Alopecia Areata |
|---|---|---|---|---|---|---|---|---|---|---|---|---|---|---|---|---|---|---|
| **PRSformer** | **0.1269** | **0.2189** | **0.0885** | **0.0224** | **0.0570** | **0.1039** | 0.0231 | **0.0163** | **0.0868** | **0.0562** | **0.0226** | **0.0328** | 0.0364 | **0.0179** | **0.0289** | 0.0898 | 0.0555 | **0.0165** |
| **LDpred2** | 0.0870 | 0.1837 | 0.0874 | 0.0194 | 0.0545 | 0.1038 | **0.0236** | 0.0143 | 0.0687 | 0.0513 | 0.0178 | 0.0324 | **0.0401** | 0.0146 | 0.0275 | **0.0927** | **0.0591** | 0.0133 |

*Note.* Predicted probabilities from both models were calibrated on the test data.

Table F12: Number of selected variants per disease across non-European ancestries

| Disease | AFR | LAT | EAS | SAS |
|---|---|---|---|---|
| Celiac | 0 | 7515 | 0 | 0 |
| T1D | 7067 | 8546 | 0 | 0 |
| Graves' | 6964 | 8889 | 6611 | 0 |
| Sjögren's | 5926 | 6645 | 0 | 0 |
| Psoriasis | 6211 | 10545 | 6524 | 6041 |
| Hashimoto Thyroiditis | 6342 | 11037 | 5780 | 0 |
| Lupus | 6172 | 7413 | 0 | 0 |
| Rheumatoid Arthritis | 6342 | 8104 | 4453 | 0 |
| Multiple Sclerosis | 5936 | 6730 | 0 | 0 |
| Crohn's | 5177 | 4959 | 0 | 0 |
| Polymyalgia Rheumatica | 0 | 0 | 0 | 0 |
| Ulcerative Colitis | 5254 | 5743 | 0 | 0 |
| Pediatric IBD | 0 | 4862 | 0 | 0 |
| Axial Spondyloarthritis | 0 | 0 | 0 | 0 |
| Vitiligo | 5599 | 7017 | 0 | 0 |
| Migratory Glossitis | 0 | 6927 | 0 | 0 |
| Canker Sore | 5732 | 6812 | 4296 | 5021 |
| Alopecia Areata | 5705 | 7521 | 4626 | 0 |

Table F13: Characteristics of multitask models trained on down-sampled datasets with subsetted input variants

| Model | # Input Variants | Train Data Size | # Model Parameters |
|---|---|---|---|
| Non-linear_downsampled | 94176 | 3000000 | 120.61M |
| Non-linear_downsampled | 38018 | 2000000 | 48.73M |
| Non-linear_downsampled | 23397 | 1500000 | 30.01M |
| Non-linear_downsampled | 15060 | 1000000 | 19.34M |
| Non-linear_downsampled | 9181 | 500000 | 11.82M |
| Non-linear_downsampled | 6829 | 300000 | 8.81M |
| Non-linear_downsampled | 4863 | 150000 | 6.29M |
| Linear_downsampled | 94176 | 3000000 | 120.55M |
| Linear_downsampled | 38018 | 2000000 | 48.66M |
| Linear_downsampled | 23397 | 1500000 | 29.95M |
| Linear_downsampled | 15060 | 1000000 | 19.28M |
| Linear_downsampled | 9181 | 500000 | 11.75M |
| Linear_downsampled | 6829 | 300000 | 8.74M |
| Linear_downsampled | 4863 | 150000 | 6.22M |

Table F14: AUROC comparison across diseases under different covariate settings (sex and age). For each disease and ancestry group, bold font denotes the best performance.

| Ancestry | Model | Covariates | Celiac | T1D | Graves' | Multiple Sclerosis | Pediatric IBD | Rheumatoid Arthritis | Vitiligo | Polymyalgia Rheumatica | Lupus | Axial Spondyloarthritis | Crohn's | Hashimoto Thyroiditis | Psoriasis | Canker Sore | Alopecia Areata | Ulcerative Colitis | Sjögren's | Migratory Glossitis |
|---|---|---|---|---|---|---|---|---|---|---|---|---|---|---|---|---|---|---|---|---|
| EUR | PRSformer | none | 0.7933 | 0.8457 | 0.7661 | 0.7184 | 0.6660 | 0.6004 | 0.6212 | 0.6048 | 0.6338 | 0.5802 | 0.6985 | 0.7504 | 0.6669 | 0.6227 | 0.6051 | 0.6442 | 0.6337 | 0.7164 |
| EUR | PRSformer | sex | **0.8051** | 0.8469 | 0.7997 | 0.7392 | 0.6707 | 0.6197 | 0.6264 | 0.6139 | 0.7199 | 0.5697 | **0.7007** | 0.8063 | 0.6684 | 0.6266 | 0.6262 | 0.6493 | 0.7292 | **0.7248** |
| EUR | PRSformer | sex+age | 0.8037 | **0.8478** | **0.8224** | **0.7575** | 0.7098 | 0.7208 | 0.6556 | 0.8536 | 0.7328 | 0.6174 | 0.7000 | **0.8123** | 0.6730 | 0.6274 | 0.6352 | 0.6757 | 0.7711 | 0.7213 |
| EUR | PRSformer-ME | none | 0.7867 | 0.8431 | 0.7674 | 0.7157 | 0.6654 | 0.6069 | 0.6222 | 0.5995 | 0.6458 | 0.5818 | 0.6981 | 0.7487 | 0.6683 | 0.6246 | 0.6125 | 0.6447 | 0.6404 | 0.7146 |
| EUR | PRSformer-ME | sex | 0.7972 | 0.8418 | 0.7996 | 0.7330 | 0.6651 | 0.6249 | 0.6254 | 0.6088 | 0.7259 | 0.5768 | 0.6991 | 0.8044 | 0.6687 | 0.6296 | 0.6347 | 0.6476 | 0.7302 | 0.7197 |
| EUR | PRSformer-ME | sex+age | 0.7993 | 0.8420 | **0.8224** | 0.7549 | **0.7124** | **0.7265** | **0.6668** | **0.8565** | **0.7425** | **0.6271** | 0.7002 | 0.8120 | **0.6758** | **0.6315** | **0.6433** | **0.6763** | **0.7766** | 0.7097 |
| AFR | PRSformer | none | 0.6219 | 0.6391 | 0.5676 | 0.5447 | 0.5928 | 0.5241 | 0.5122 | – | 0.5404 | <0.5 | 0.5726 | 0.6727 | 0.5559 | 0.5913 | <0.5 | 0.5401 | 0.5118 | 0.6008 |
| AFR | PRSformer | sex | 0.6346 | 0.6496 | 0.6153 | 0.5709 | 0.5983 | 0.5630 | 0.5207 | – | 0.6500 | <0.5 | 0.5755 | 0.7473 | 0.5674 | 0.5932 | <0.5 | 0.5554 | 0.6068 | 0.6054 |
| AFR | PRSformer | sex+age | 0.6412 | 0.6589 | 0.6964 | 0.5997 | 0.6161 | 0.7445 | 0.5699 | – | 0.6917 | <0.5 | 0.5790 | 0.7695 | 0.5797 | 0.5968 | 0.5322 | 0.5971 | 0.6923 | 0.6110 |
| AFR | PRSformer-ME | none | 0.6269 | 0.6986 | 0.6997 | 0.6397 | 0.6285 | 0.5660 | 0.5605 | – | 0.6140 | 0.5410 | **0.6217** | 0.6939 | 0.5822 | 0.6026 | 0.5844 | 0.5902 | 0.6398 | 0.6550 |
| AFR | PRSformer-ME | sex | **0.6576** | 0.7046 | 0.7431 | 0.6834 | 0.6215 | 0.5971 | 0.5779 | – | 0.7007 | 0.5460 | 0.6145 | 0.7559 | 0.5872 | 0.6070 | 0.6243 | 0.6021 | 0.7026 | **0.6577** |
| AFR | PRSformer-ME | sex+age | 0.6446 | **0.7092** | **0.7893** | **0.7150** | **0.6647** | **0.7662** | **0.6350** | – | **0.7346** | **0.6374** | **0.6217** | **0.7782** | **0.5972** | **0.6108** | **0.6563** | **0.6494** | **0.7688** | 0.6463 |
| LAT | PRSformer | none | 0.7437 | 0.7459 | 0.7060 | 0.6832 | 0.6216 | 0.5712 | 0.5871 | 0.6526 | 0.6063 | 0.5610 | 0.6674 | 0.7482 | 0.6449 | 0.6283 | 0.5579 | 0.6184 | 0.6181 | 0.6994 |
| LAT | PRSformer | sex | 0.7599 | 0.7491 | 0.7486 | 0.7055 | 0.6261 | 0.6042 | 0.5882 | 0.6641 | 0.6930 | 0.5689 | 0.6674 | 0.8000 | 0.6512 | 0.6299 | 0.5751 | 0.6262 | 0.6982 | 0.7088 |
| LAT | PRSformer | sex+age | 0.7573 | 0.7580 | 0.7875 | 0.7316 | 0.6490 | 0.7516 | 0.6226 | 0.8390 | 0.7225 | 0.6270 | 0.6697 | 0.8135 | 0.6601 | 0.6299 | 0.5985 | 0.6661 | 0.7613 | 0.7083 |
| LAT | PRSformer-ME | none | 0.7521 | 0.7732 | 0.7568 | 0.7256 | 0.6539 | 0.5979 | 0.6166 | 0.6553 | 0.6536 | 0.6038 | 0.6842 | 0.7581 | 0.6596 | 0.6397 | 0.6346 | 0.6291 | 0.6685 | 0.7270 |
| LAT | PRSformer-ME | sex | **0.7649** | 0.7762 | 0.7915 | 0.7427 | 0.6548 | 0.6300 | 0.6210 | 0.6756 | 0.7335 | 0.5962 | 0.6827 | 0.8081 | 0.6601 | 0.6417 | 0.6387 | 0.6294 | 0.7382 | **0.7279** |
| LAT | PRSformer-ME | sex+age | 0.7642 | **0.7792** | **0.8170** | **0.7658** | **0.6819** | **0.7671** | **0.6601** | **0.8588** | **0.7616** | **0.6700** | **0.6900** | **0.8222** | **0.6713** | **0.6426** | **0.6522** | **0.6736** | **0.7962** | 0.7241 |
| SAS | PRSformer | none | 0.8180 | 0.6640 | 0.7667 | 0.7341 | 0.6423 | 0.6052 | 0.5693 | – | 0.5656 | 0.5999 | 0.6630 | 0.6705 | 0.6209 | 0.5999 | 0.5754 | **0.6974** | 0.6383 | 0.5421 |
| SAS | PRSformer | sex | **0.8183** | 0.6550 | 0.7756 | 0.7496 | 0.6330 | 0.6455 | 0.5805 | – | 0.6754 | 0.5857 | 0.6618 | 0.7377 | 0.6224 | 0.6053 | 0.5843 | 0.6865 | 0.7393 | **0.5915** |
| SAS | PRSformer | sex+age | 0.8042 | 0.6685 | 0.8185 | 0.7536 | 0.7044 | 0.7479 | 0.5993 | – | 0.6860 | 0.6193 | 0.6684 | 0.7479 | 0.6186 | **0.6146** | 0.5867 | 0.6862 | 0.7593 | 0.5706 |
| SAS | PRSformer-ME | none | 0.8002 | **0.6852** | 0.7612 | 0.7503 | 0.6858 | 0.5974 | 0.6009 | – | 0.6196 | 0.5828 | **0.7061** | 0.6907 | 0.6269 | 0.5813 | 0.5972 | 0.6969 | 0.6885 | 0.5146 |
| SAS | PRSformer-ME | sex | 0.8108 | 0.6743 | 0.7784 | **0.7813** | 0.6663 | 0.6666 | 0.5935 | – | 0.7053 | 0.6221 | 0.6732 | 0.7452 | 0.6230 | 0.6069 | 0.6042 | 0.6653 | 0.7423 | 0.5492 |
| SAS | PRSformer-ME | sex+age | 0.8096 | 0.6796 | **0.8231** | 0.7737 | **0.7561** | **0.7632** | **0.6104** | – | **0.7097** | **0.6559** | 0.6748 | **0.7580** | **0.6325** | 0.6032 | **0.6041** | 0.6861 | **0.7701** | 0.5348 |
| EAS | PRSformer | none | – | 0.6507 | 0.6937 | 0.5406 | 0.5902 | 0.5566 | 0.5463 | 0.7431 | 0.6144 | 0.5184 | **0.6282** | 0.6535 | 0.6167 | 0.5731 | 0.5846 | 0.6204 | 0.5770 | 0.6809 |
| EAS | PRSformer | sex | – | 0.6357 | 0.7357 | 0.6136 | 0.6106 | 0.5808 | 0.5425 | 0.7208 | 0.6893 | <0.5 | 0.5928 | 0.7335 | 0.6182 | 0.5642 | 0.5986 | 0.6027 | 0.6541 | 0.6659 |
| EAS | PRSformer | sex+age | – | 0.6588 | 0.7585 | **0.6612** | 0.6153 | 0.7759 | 0.5902 | **0.8856** | 0.7164 | 0.5640 | 0.5932 | 0.7491 | 0.6298 | 0.5773 | 0.6154 | 0.6607 | 0.7467 | 0.6860 |
| EAS | PRSformer-ME | none | – | 0.7121 | 0.7487 | 0.5019 | **0.6518** | 0.6085 | 0.5602 | 0.6973 | 0.6573 | 0.5860 | 0.6256 | 0.6876 | 0.6355 | 0.5929 | 0.6386 | 0.6545 | 0.6176 | **0.7085** |
| EAS | PRSformer-ME | sex | – | 0.7001 | 0.7828 | 0.5988 | 0.6385 | 0.6380 | 0.5725 | 0.6981 | 0.7295 | 0.6081 | 0.5960 | 0.7518 | 0.6331 | **0.5973** | 0.6379 | 0.6345 | 0.6918 | 0.6938 |
| EAS | PRSformer-ME | sex+age | – | **0.7157** | **0.7964** | 0.6455 | 0.6222 | **0.7978** | **0.6193** | 0.8587 | **0.7450** | **0.6586** | 0.6001 | **0.7729** | **0.6449** | 0.5972 | **0.6662** | **0.6980** | **0.7742** | 0.7041 |

