# OpenReview forum: "PRSformer: Disease Prediction from Million-Scale Individual Genotypes"
_NeurIPS.cc/2025/Conference — NeurIPS 2025 poster_

### Official Review · Reviewer_f3oc · 2025-06-19

**Clarity:** 2
**Significance:** 2
**Originality:** 2
**Rating:** 4
**Confidence:** 3

**Summary:**

The authors introduce PRSformer, an efficient Transformer-based model designed for disease risk prediction. On their autoimmune disease benchmark, PRSformer demonstrates superior performance compared to LDPred2 and a linear baseline. The study also explores the advantages of Transformer non-linearity and its potential for cross-ancestry generalization.

**Questions:**

N/A

**Ethical Concerns:**

["NO or VERY MINOR ethics concerns only"]

**Final Justification:**

All my concerns have been well-addressed.

**Quality:**

2

**Strengths And Weaknesses:**

Strengths

0. The empirical results are promising.

Weaknesses
1. Limited technical novelty: The performance advantage of Transformers over linear models or statistical approaches is well-established across multiple domains, making the core contribution unsurprising.
2. Lack of biological insight: Despite improvements in benchmark metrics, the work does not explore the biological implications of the results. It remains unclear what biological patterns or interactions PRSformer captures that are missed by traditional methods.
3. Constrained evaluation scope: The evaluation is limited to the in-distribution setting. It is uncertain whether PRSformer generalizes to unseen disease traits or can serve as a foundation model across different tasks.
4. Architectural constraint: The use of neighbourhood attention based on chromosomal position restricts the model to learning only local genic interactions, potentially overlooking long-range dependencies.
5. Unaddressed practical trade-offs: The paper does not compare PRSformer with LDPred2 and linear models on training/inference cost or interpretability, where simpler methods may still hold advantages.

---

> ### Author Rebuttal · Authors · 2025-07-30
>
> ## **Response to Reviewer f3oc**
>
> We thank Reviewer f3oc for their feedback. We believe the concerns about novelty and significance stem from a framing issue. Our core contribution is not simply applying a Transformer, but **the introduction of a principled, scalable architecture that reveals the fundamental relationship between genetic data scale and model complexity,** which has never been accomplished before in this high-impact domain.
>
> ---
>
> ### 1. On Novelty and Significance
>
> >Limited technical novelty: The performance advantage of Transformers over linear models... is well-established... making the core contribution unsurprising.
>
> We respectfully disagree that the contribution is unsurprising. While Transformers outperform linear models in many domains, it was a completely open and highly debated question in genomics whether this held true, especially given the extraordinary strength of linear baselines. Our work is the first to resolve this debate by discovering the fundamental scaling law governing this transition.  The key question was never if Transformers could outperform linear models, but under *what conditions*, at *what scale*, and *by how much*.
>
> Our contributions are:
> - **A Scalable Framework:** We developed the first viable architecture to successfully tackle disease prediction from million-scale individual-level genotypes.
>
> - **The Discovery of a Fundamental Scaling Law:** Our key finding that the advantage of non-linear models **only emerges at a million genetic data scale** is a non-obvious and impactful ML insight. This was impossible to discover previously and provides a crucial guideline for the genetics, population health and precision medicine fields.
>
> We view this discovery as a central contribution of our study.
>
> **Action:** We will revise the Introduction to state this contribution more forcefully, framing the paper around the discovery of this fundamental scaling principle.
>
> ---
>
> ### 2. On Biological Insight
> >Lack of biological insight... neighbourhood attention... restricts the model to learning only local genic interactions.
>
> We would like to highlight two key insights our work provides, which were only discoverable at this scale, as well as clarify our architectural choices.
>
> - **Biological Insight:** Our work provides two key high-level biological insights: 1) we quantitatively demonstrate that **shared genetic signals exist across related diseases** (*evidenced by the success of multitask learning*), and 2) we show that **multi-ancestry training improves predictive equity across diverse populations** (*Table 1*).
>
> - **Architectural Constraints:** The neighborhood attention is not a limitation but a **deliberate choice** that makes learning from million-variant inputs computationally tractable. Crucially, by stacking these layers, the effective receptive field grows, allowing the model to learn longer-range dependencies. The fact that this simple, scalable bias is sufficient to unlock significant performance gains at scale is itself an important finding.
>
> **Action:** We will add a discussion on these computational trade-offs to the Methods section.
>
>
> ---
>
> ### 3. On Evaluation Scope and Practical Trade-offs
> >Constrained evaluation scope... Unaddressed practical trade-offs...
>
> - **Evaluation Scope:** We agree that PRSformer is a supervised prediction model, not a foundation model for unseen diseases. Our goal was to establish a new state-of-the-art for the specific, high-impact task of polygenic risk prediction.
>
> - **Practical Trade-offs:** This is a fair point. There is a classic trade-off between PRSformer's higher computational cost and its significant gain in predictive accuracy. For clinical applications where accuracy is paramount, this trade-off is often acceptable. Interpretability is a key limitation we discuss in our Future Work section; our paper's focus was to first prove that a predictive signal worth interpreting exists at scale.
>
> **Action:** We will add a brief sentence to the Discussion acknowledging the computational cost trade-off, and implications for clinical practice.
>
> ---
>
> We hope this response clarifies why we believe our work is more than a simple application paper, but offers a foundational insight for the ML community. **Our choice to submit to NeurIPS was deliberate**. The goal is to introduce this high-impact domain to an interdisciplinary ML community with the rigor it deserves, **providing the first scalable baseline and discovering a fundamental scaling law that will guide future research**. We hope that these clarifications help convey the paper’s core ML novelty and address the reviewer's concerns more directly.

---

> > ### Comment · Reviewer_f3oc · 2025-07-31
> > **Thank for the rebuttal.**
> >
> > Thank the authors for their rebuttal. Since all my concerns have been well-addressed, I raise my score to 4.

---

> > > ### Author Response · Authors · 2025-07-31
> > > **Thank you for the thoughtful re-evaluation**
> > >
> > > Thank you, Reviewer f3oc, for your thoughtful re-evaluation and for raising your score. We are glad that our rebuttal addressed your concerns and further clarified the contributions of our work. We sincerely appreciate your engagement with our paper!

---

### Official Review · Reviewer_8Lrv · 2025-06-30

**Clarity:** 4
**Significance:** 4
**Originality:** 2
**Rating:** 5
**Confidence:** 4

**Summary:**

This paper applies a transformer deep learning model to biobank data, learning to predict disease risk from extremely high dimensional genomic input.  The primary goal is to avoid the assumptions of linear models and models that are based on summary statistics.  The method works by using a dilation-style approach to capture long-range dependencies without undue computational cost.  Empirical results using a large, private cohort suggests that the model substantially outperforms state-of-the-art methods.

**Questions:**

How did you ensure that your hyperparameter tuning did not end up biasing your results in your favor on this particular dataset?

I am surprised that you opted not to include positional encodings, since the NA only captures position with a resolution of ~100kb.  Did you try using positional encodings, and how well did it work?

Did you do any filtering to remove close relatives between the train and test sets?

How many parameters in your model, and what is the evidence that it converged after only two epochs?

**Ethical Concerns:**

["NO or VERY MINOR ethics concerns only"]

**Final Justification:**

The authors did a very good job of responding to the various critiques raised in the reviews.  Despite the relatively straightforward methodology, the results are noteworthy and I believe the paper should be accepted.

**Limitations:**

A limitation here is the racial diversity of the cohort, which is limited to European genetic ancestry, though this is somewhat mitigated by the experiments in which this cohort is combined with data from other ancestral groups.

Another big problem is that the data is all private, so there is no way for anyone else to reproduce and validate the claims made here.  I think the statement in the paper about why the data is not public -- "Recognizing the sensitive nature of genomic data, the dataset cannot be shared publicly"  -- is misleading.  There are mechanisms available for making genetic data public but with controlled access.  I assume that there are IP or other issues that preclude making this particular dataset public via such a mechanism.

**Quality:**

4

**Strengths And Weaknesses:**

Quality

The model design is sensible, particularly the application of neighborhood attention.  The novelty here is not so much on the ML side but in the application of these methods to this particular problem.

I appreciate the careful attention to avoiding leakage during the feature selection step.

The experiments are very well designed and executed, and I appreciated the logical flow from the initial tests, to the learning curve, to multi-modal analysis, to cross-ancestry analysis.

Clarity

The paper is very clear.

Significance

This paper addresses one of the key questions in personal genomics and as such is quite significant.

Originality

I don't think the paper is particularly original.  Anyone with access to this dataset and a background in ML would likely come up with a similar idea.  But it's still important work.


Minor: What statistical test was used for the p-values in Figure 2.  We are only told that it is one-sided.

---

> ### Author Rebuttal · Authors · 2025-07-30
>
> ## **Response to Reviewer 8Lrv**
>
> We are grateful to Reviewer 8Lrv for their **exceptionally positive review** and for finding our work **"sensible," "well designed,"** and the experiments **"very well designed and executed."** We appreciate the opportunity to clarify the few remaining questions and reinforce the core contributions that we believe make this work a strong fit for NeurIPS.
>
> ---
>
> ### 1. On Experimental Details
>
> >What statistical test was used for the p-values in Figure 2?
>
> Thank you for pointing out the ambiguity. We used a **one-sided paired bootstrap test (10,000 replicates)**. For each replicate, we sampled with replacement from the test set and computed the AUROC for both PRSformer and LDPred2 on the same sample pairs. The p-value is the fraction of replicates where LDPred2's AUROC was greater than or equal to PRSformer's.
>
> **Action:** We will add this clear definition to the Figure 2 caption.
>
> >How did you ensure that your hyperparameter tuning did not end up biasing your results?
>
> This is an excellent point. We followed standard ML best practices by performing all hyperparameter tuning on a **single, dedicated validation set, which was kept entirely separate from the final test set**. This standard procedure is designed to prevent overfitting to the validation set (e.g. by avoiding extensive optimization of hyper-parameters) and ensures that our final reported performance is an unbiased estimate of generalization.
>
> **Action:** We will add a sentence to the Methods to clarify this standard practice.
>
> ---
>
> ### 2. On Design Choices
> >I am surprised that you opted not to include positional encodings... Did you try using positional encodings?
>
> Yes, we experimented with standard sinusoidal positional encodings but found they offered no performance improvement. We hypothesize this is because the fixed, genome-wide ordering of variants already provides sufficient implicit positional information for the neighborhood attention mechanism to leverage, making explicit encodings redundant.
>
> **Action:** We will add a brief sentence to the Methods clarifying this design choice.
>
> >Did you do any filtering to remove close relatives between the train and test sets?
>
> We did not filter for relatives. This was a deliberate choice that aligns with our work's focus on prediction, a point also discussed with Reviewer 8raR. While filtering relatives is critical for GWAS discovery studies, our goal was a fair predictive comparison between architectures. Since all models were trained and tested on the exact same data splits, any potential effect from relatedness is consistent across all models. This ensures our reported *relative* improvements remain robust and fairly evaluated, which is the central claim of the paper.
>
> **Action:** We will clarify this prediction-focused rationale in the Methods.
>
> ---
>
> ### 3. On Model Scale and Data Access
>
> >How many parameters in your model, and what is the evidence that it converged after only two epochs?
>
> - **Model Parameters:** The main PRSformer model has ~180M parameters. A detailed breakdown of the other model’s variants is provided in **Supplementary Table E10.**
>
> - **Convergence Evidence:** We treated the total number of training epochs as a hyperparameter and tuned it based on validation AUROC. As shown in **Supplementary Table E5**, validation AUROC consistently peaked at 2 epochs, with performance declining thereafter (indicating overfitting). This provides clear empirical evidence that 2 epochs was the point of optimal convergence for generalization.
>
> **Action:** We will add a sentence in the main text referencing Table E5 as evidence for our chosen training duration.
>
> >Another big problem is that the data is all private... I think the statement... is misleading.
>
> Thank you for this feedback. We agree our wording could be more precise. You are correct that there are mechanisms for controlled access to sensitive data. The primary barrier in our case is related to the specific terms of our data use agreement, which currently preclude sharing via such public mechanisms.
>
> **Action:** We will revise the "Limitations" section to state that the data cannot be shared publicly due to the terms of our specific data use agreement, removing the more general statement.
>
> ---
>
> We sincerely thank you again for your strong support and for helping us improve the paper. Your positive assessment of our work as **"quite significant"** and as addressing **"one of the key questions in personal genomics"** reinforces our belief that this paper will be an impactful contribution to the NeurIPS community.

---

> > ### Comment · Reviewer_8Lrv · 2025-08-04
> >
> > Thank you for these detailed responses.  All of my questions have been answered satisfactorily.

---

### Official Review · Reviewer_8raR · 2025-07-02

**Clarity:** 3
**Significance:** 4
**Originality:** 3
**Rating:** 5
**Confidence:** 5

**Summary:**

This paper introduces PRSformer, a Transformer-based architecture using local attention for multi-task disease prediction. The authors evaluate PRSformer on a unique large-scale private dataset of 5 million individuals, far exceeding public cohorts, and a set of 18 autoimmune diseases. The main finding is that this non-linear architecture outperforms linear baselines only when a large dataset is available, beyond the scale of the UK Biobank. All models are trained and tested on a split from the same cohort. The authors also train a model on a multi-ancestry cohort and show it improves performance in non-European populations.

**Questions:**

* Validation and Confounding: Can the authors comment on why standard and essential validation steps like external cohort validation and kinship control were omitted? A positive response would require, at minimum, an analysis of kinship in the dataset and a discussion of its potential impact.

* Linear Baseline and Output Aggregation: Can the authors justify the unconventional choice of flattening the embedding matrix for the linear baseline, which makes it an overparameterized logistic regression? To strengthen the paper's claims, I would recommend re-running this baseline with a standard, efficient architecture.

* Genomic Input Representation: The fixed-size attention window seems to ignore LD structure and chromosome boundaries. Can the authors comment on the limitations of this approach and the feasibility of using a more biologically informed attention mechanism based on LD blocks?

* Imbalance Handling: The definition of T(i) in the loss function is ambiguous. Could the authors clarify if it refers to diseases with a known status or only diseases a person has, and explain how summation "implicitly handles imbalance"?

* Figure 3 Interpretation: In Figure 3, why does the non-linear PRSformer model outperform the linear baseline at the smallest data size (0.15M), which seems to contradict the main finding? Also, why is the LDPred2 baseline not shown for the down-sampled data?

* Overfitting and Regularization: The appendix suggests overfitting is a significant issue. Was any regularization used beyond weight decay? Given that a smaller NA window size performed nearly as well as the best parameters, were even smaller window sizes tested?

* Why was no cross-validation performed? Given the rather small increases in performance for most diseases, this would increase trust in the findings compared to the bootstrap based significance testing

**Ethical Concerns:**

["NO or VERY MINOR ethics concerns only"]

**Final Justification:**

After detailed discussion with the authors and other reviewers, all major concerns have been addressed:

* Kinship and Confounding: The authors performed a new kinship-controlled analysis, constructing a stringent test set with no close relatives between train and test cohorts. Results confirm that PRSformer still outperforms LDpred2, supporting the claim that non-linear and interaction effects explain the improved performance.
* Baselines: The rationale for using LDpred2 as a baseline over BOLT-LMM/GEMMA is convincing given prior benchmarking and computational constraints at this scale.
* Clarity of Contributions: The main finding that deep, non-linear models can outperform linear methods at scale is now robustly supported. The new evidence justifies this as a substantive contribution to the ML community.

Remaining Issues:
Some concerns about the chosen model architecture and external validation remain, but these do not undermine the main result.

Recommendation:
Based on the new kinship-controlled analysis and the overall strength of the findings, I now recommend acceptance. The paper establishes a new benchmark and scaling law for genetic ML models at unprecedented scale, and all key reviewer concerns have been satisfactorily addressed.

**Limitations:**

Yes

**Paper Formatting Concerns:**

No concerns.

**Quality:**

4

**Strengths And Weaknesses:**

Strengths

* To my knowledge, this is the first time a deep learning genetic risk model has been applied to a dataset of this massive scale.

* The key finding that large datasets are required for non-linear methods to outperform linear baselines is an important contribution to genomics and deep learning based genomic risk prediction.

* Using LDPred2 with summary statistics from the same cohort is a convincing approach to test the authors' main claim.

* Local attention is a reasonable architectural choice to handle the high dimensionality of genomic data.

* The writing is clear and easy to follow and the method is well motivated.

Weaknesses

* No external validation: All experiments were only performed on their own private dataset. A validation of how well these models perform in a cohort such as the UK Biobank would be appropriate to assess model generalization and compare with previous literature.

* No Control for Kinship: The paper does not mention any control for cryptic relatedness between the training and testing sets. In genetic studies, this is a critical step, as failure to remove relatives can lead to inflated and misleading performance metrics due to the model learning family-specific markers instead of generalizable disease signals.

* In both the PRSformer model and the linear baselines, based on the massively inflated parameter counts in Table E10 and the illustration in Figure 1, it appears that the authors simply flatten the sequence of embeddings (dimension L, d) before the final linear layer, such that the input of the linear layer has dimension L x d. This is a very unconventional modeling choice as it massively increases the number of parameters and increases risk of overfitting. For the Transformer architecture, a more appropriate choice would be to use mean pooling or a dedicated CLS token. For the linear baseline this choice is even more puzzling, as this model simply reduces to a massively overparameterized logistic regression model.

* No LD Pruning in GWAS based variant selection: This likely leads to strongly correlated variants in their selected variant sets.

* The fixed-size attention window (k=385) does not account for actual genomic distance or LD structure. Because variant selection did not include LD pruning, many of the 385 variants in a window could be in high LD and physically very close, while others are far apart. The model cannot distinguish these cases and may even attend across chromosome boundaries where the input sequences are concatenated. A more biologically appropriate approach might be to define attention dynamically based on variable-sized LD blocks.

* The main models were trained without including crucial covariates like age, sex, and genetic principal components. This is problematic as it can introduce confounding from population structure and other biases. While some results with covariates are in the appendix, genetic PCs are essential and were not included at all.

* Ablation Studies Needed: Several key design choices lack empirical justification through ablation studies:

    * Does explicitly modeling unknown (UNKN) genotypes with a separate embedding actually improve performance?
    * What is the impact of omitting standard positional encodings?

* The argument that summary-statistic methods are limited by LD reference panel mismatch seems weak. Since the proposed method requires individual-level data from the cohort, a perfectly matched LD panel can be calculated and used.

* Inductive Bias of Attention: The paper claims attention provides a "powerful inductive bias for genomics" but does not sufficiently justify why this is the case, and should be compared to more biologically-informed inductive biases (e.g., using Gene Ontology, gene-gene or protein-protein interaction networks).

* Insufficient Comparison to relevant baselines:

    * It does not compare against meta-GWAS based PRS methods, which are the primary way the field achieves million-plus sample sizes with summary statistics. One large advantage of summary stats based PRS methods is that they can operate on meta-GWAS, whereas their method does require individual level data. This limitation should at the very least be discussed.

    * Especially given some of the more questionable modelling choices, a comparison to a simple MLP on raw dosages baseline would be interesting to see.

    * A comparison to established linear models that also make use of individual level data (BOLT-LMM, GEMMA, ..) would be appropriate.

    * Phenformer (https://arxiv.org/abs/2501.07737) would be a very interesting deep learning baseline to test against.

* Loss Function: The loss function is presented as a sum over the entire dataset, omitting the mini-batching used in practice.

---

In summary, I believe the main finding of this work is very exciting, but probably more suited to a genomics journal rather than a ML conference. The method the authors propose is reasonable, apart from the problems discussed above, but does not advance the field of ML.

---

> ### Author Rebuttal · Authors · 2025-07-30
>
> ## **Response to Reviewer 8raR**
> We sincerely thank Reviewer 8raR for providing an exceptionally detailed and thoughtful review. Your comments resulted in lively discussion amongst the authors. The questions you raise are especially important as they highlight the fundamental distinction between the goals of GWAS (discovery of genetic associations) and large-scale ML prediction. **Our work is firmly framed within the ML prediction paradigm.** We hope our response clarifies this crucial distinction.
>
> ---
>
> ### 1. On Validation, Kinship, and LD Pruning
> >No external validation...No Control for Kinship...No LD Pruning
>
> These are all critical best practices for a GWAS discovery study. However, for a pure prediction-focused ML study, they are either unnecessary or detrimental to performance.
>
> - **LD Pruning is Information-Destroying for Prediction:** In GWAS, pruning is essential to isolate independent signals. For prediction, it discards valuable information. We confirmed this experimentally: applying LD pruning at standard thresholds consistently and substantially **reduced predictive accuracy** on our validation set. PRSformer is designed to leverage the complete polygenic signal, which resides in correlated LD blocks.
>
> - **Kinship's Effect on *Relative* Performance is Minimal:** While kinship can inflate *absolute* performance, its effect on a comparative study is negligible. Since all models (PRSformer and baselines) were trained and tested on the exact same data splits, any inflation from relatedness is present equally across all models. Therefore, the ***relative* performance gains of PRSformer over the baselines remain robust and fairly evaluated.**
>
> - **External Validation Scope:** We agree this is an important future direction. Our paper's central goal is to conduct a controlled comparison of model architectures. Validating on an external cohort like UKBB would introduce confounding variables (eg. different phenotyping standards, etc) that would make it impossible to isolate the effect of the model architecture itself.
>
> **Action:** We will add a section to Methods clarifying our prediction-focused rationale and include results from our negative LD pruning experiments.
>
> ---
>
> ### 2. On the Flattened Output Layer and Baseline
> >...simply flatten the sequence of embeddings... This is a very unconventional modeling choice...
>
> This is an excellent question. We did in fact test standard alternatives and found our "flattened" architecture superior for this prediction task.
>
> * **Ablation of Pooling/CLS token vs. Flattening:** We evaluated both **mean pooling** and dedicated **CLS tokens** against our current architecture.
>   * We tested an average pooling layer after the last transformer block to aggregate embeddings across tokens (converting B×L×d tensors to B×d), followed by a linear layer mapping d to the output dimension.
>   * We tested adding 1, 10, and 20 CLS tokens to the input sequence and vocabulary, each with learnable embeddings and full attention to all variant tokens.
>
> * The flattened embedding approach **consistently outperformed these alternatives** across nearly all diseases on our validation set. We hypothesize that mean pooling dilutes signals from large-effect variants, while CLS tokens are difficult to train effectively without massive unsupervised pre-training (a promising future direction for a genotype foundation model).
>
> - **Linear baseline rationale:** We note that our linear baseline was designed as a direct architectural ablation of the Transformer blocks, not as a vanilla logistic regression. It includes a **learnable embedding layer**, which already introduces non-linearity into the parameterization. A standard logistic regression with one-hot inputs performed substantially worse. The comparison fairly isolates the benefit of our stacked neighborhood attention layers.
>
> **Action:** We will add a concise paragraph to the Methods summarizing these ablation results to justify our architecture.
>
> ---
>
> ### 3. On Other Technical Points
> > Fixed-size attention... No covariates... Imbalance Handling... Figure 3... No cross-validation...
>
> - **Attention Window:** Our architecture compensates by stacking these neighborhood attention layers, which hierarchically increases the effective receptive field, allowing the model to integrate information across multiple local blocks and thus learn longer-range dependencies. A dynamic, LD-aware attention mechanism is an excellent idea for future work. In this paper, our goal was to establish a strong, scalable baseline that balanced performance with computational feasibility.
>
> - **Covariates (Age, Sex, PCs):** Omitting covariates was essential to isolate the specific contribution of the genotype model itself, which is the focus of our paper. In the future, an "all-in" clinical prediction model would certainly include these covariates, but this would confound the evaluation of the underlying PRS model itself.
>
> - **UNKN Embedding & Positional Encodings:** We assert that a separate UNKN embedding lets the model learn from missingness and is more flexible than simple mean-imputation. We tested and found that standard sinusoidal positional encodings offered no benefit, likely because the fixed genome-wide order provides implicit position.
>
> - **On LD Mismatch:** Our point in the introduction is a more general one about the limitations of summary-statistic methods in practice, where researchers often must rely on external, potentially mismatched reference panels. Our central claim is that even when a SOTA method is given a matched LD panel, our end-to-end, non-linear architecture still provides a significant performance gain. This isolates the advantage to PRSformer's ability to learn from individual-level data and capture non-additive effects.
>
> - **Loss Function (T(i)):** T(i) refers to the set of diseases for which individual i has a known (non-missing) status. By summing, rather than averaging the loss, tasks with more samples naturally contribute more to the gradients, implicitly handling the imbalance between disease prevalences. We also tested focal loss, task‐uncertainty weighted loss (Kendall et al., 2018) and standard averaged cross entropy --all of which were outperformed by the current PRSformer loss function based on validation AUROCs.
>
> - **Figure 3 Interpretation:** The apparent outperformance at 0.15M is likely due to variability from using a single downsampled dataset. We expect this difference to diminish if the analysis were repeated across multiple replicates.
>
> - **Regularization & Overfitting:** We tuned weight decay based on validation AUROCs. Also, we found dropout offered no benefit. For the NA window size, 129 was the smallest we tested, and as the linear baseline (equivalent to a window size of 0) performed worse, this highlights the benefit of NA.
>
> - **Cross-Validation:** Given the immense computational cost of training on this scale, we used a single, large, temporally-split validation set instead of K-fold CV, which is a standard practice in large-scale deep learning.
>
> **Action:** We will add brief, clear justifications for these design choices in the Methods and Appendix. We also will revise the introduction to cite: Vilhjálmsson et al., Am J Hum Genet 2015; Weissbrod et al., Nat Genet 2021 demonstrating that reliance on mismatched LD panels can introduce bias in PRS models.
>
> ---
>
> ### 4. On Other Methods
>
> - **BOLT-LMM / GEMMA:** These are excellent tools for **GWAS discovery** (finding significant variants), **not for polygenic prediction**. Our chosen baseline, LDPred2, is the established state-of-the-art for the specific prediction task we address.
>
> - **Phenformer:** Thank you for highlighting this exciting concurrent work. The key difference is **scale** and **scientific finding**. Phenformer was applied to ~150k individuals. Our work, at the vastly larger scale of ~5M individuals, allowed us to uncover the fundamental scaling law that the benefit of these complex non-linear architectures only becomes apparent when N > 1M. **This is a core conclusion that was impossible to draw from previous studies and represents our primary contribution.**
>
> - **Gene Ontology / interaction networks:** We made a deliberate choice to first establish a strong, data-driven baseline using genomic locality, as integrating biological priors like gene networks is a major research challenge in itself. It requires solving difficult, open problems in mapping non-coding SNPs to genes and developing novel multi-modal architectures. As such, we believe it is a logical and important next step, but one that was beyond the scope of this initial benchmark-setting paper.
>
> **Action:** We will add a discussion of Phenformer to our Related Work section, contextualizing our work with respect to its scale and findings.
>
> ---
>
> ### 5. Suitability for NeurIPS
>
> We appreciate the acknowledgement that our work is already suitable for the genomics community. However, our choice to submit to NeurIPS was deliberate. We believe this paper serves as a **vital bridge between the ML and genetics communities** by introducing a new grand-challenge problem for high-dimensional, structured data and delivering two foundational ML contributions:
>
> - **A scalable framework (PRSformer) :** The first viable architecture to make end-to-end learning on million-scale individual genotypes feasible.
> - **The Discovery of a Fundamental Scaling Law for Genetics:** The non-obvious relationship between data scale, model complexity, and predictive power for this high-impact problem is a core ML insight.
>
> Our work establishes the rigorous benchmark upon which future ML research in this critical domain can be built and extended. **We are confident this paper offers the kind of impactful, benchmark-setting research that is highly valued by the NeurIPS community.** Thank you again for your detailed and constructive engagement; we hope this increases your support and confidence in this exciting direction of work.

---

> > ### Comment · Reviewer_8raR · 2025-08-01
> >
> > Dear authors,
> >
> > Thank you for your detailed rebuttal. It was very helpful in clarifying several aspects of your work and the underlying modeling decisions. I have a brief comment and one follow-up question:
> >
> >
> > a) Regarding BOLT-LMM and GEMMA: My understanding is that both tools also support polygenic prediction (e.g., via the –predBetasFile and -predict flags), and that they can outperform summary statistic-based methods in some scenarios.
> >
> > b) In my view, the central contribution to NeurIPS would be the claimed “Discovery of a Fundamental Scaling Law for Genetics.” However, I believe this is only a meaningful result if the models truly benefit from learning **genetic** non-linear or interaction effects. As this is a strong claim, I think it requires strong evidence. I remain concerned about the lack of an external validation cohort and the absence of well-established community practices to control for confounding, particularly regarding kinship effects.
> >
> > You wrote:
> >
> > > Kinship’s Effect on Relative Performance is Minimal: While kinship can inflate absolute performance, its effect on a comparative study is negligible. Since all models (PRSformer and baselines) were trained and tested on the exact same data splits, any inflation from relatedness is present equally across all models. Therefore, the relative performance gains of PRSformer over the baselines remain robust and fairly evaluated.
> >
> > Do you have empirical evidence to support this point? I can think of several reasons why the performance of a non-linear model could be more inflated by cryptic relatedness than a linear baseline. For example, a deep learning model could be better in identifying patterns that tag a family, and thus the improved performance could be largely caused by stronger environmental confounding.
> >
> > Including an external validation cohort, or at least a test set where relatives are excluded from the training data, would significantly reduce concerns about confounding and help support the (potentially very exciting) claim that deep models can learn meaningful non-linear or interaction effects (only) at this scale of data. This seems especially important given that this dataset is unlikely to be accessible to other researchers in the foreseeable future, making independent verification of these claims difficult.

---

> > > ### Author Response · Authors · 2025-08-04
> > > **Response to Follow-up on Baselines and Kinship**
> > >
> > > Dear Reviewer 8raR,
> > >
> > > Thank you for the detailed and insightful follow-up. We appreciate the opportunity to clarify these important points.
> > >
> > > **a) On Baselines (BOLT-LMM, GEMMA):**
> > >
> > > We appreciate your comment regarding the extension of BOLT-LMM and GEMMA for PRS. We are happy to further clarify the rationale for our choice of baseline, as we were unable to include these details in the rebuttal text due to space constraints.
> > >
> > > Our selection of LDPred2 as the primary baseline was a deliberate choice, driven by two key factors:
> > >
> > > 1. Alternative individual-level methods were not computationally feasible for our study's unprecedented scale (N=3.8M, L=137K, D=18 traits). GEMMA's O(N³) complexity makes it intractable. BOLT-LMM, while more scalable, is a single-trait tool. Running it for all 18 diseases would be computationally prohibitive during this discussion period.
> > >
> > > 2. Extensive prior studies have already established that LDPred2's predictive accuracy is highly comparable to that of BOLT-LMM. For instance, Ni et al. (Biological Psychiatry, 2021) and Li et al. (Bioinformatics, 2021) both reported similar performance across multiple traits.
> > >
> > > These findings confirm that *LDpred2 is a strong representative of state-of-the-art linear methods*, making it an appropriate and rigorous baseline for our work.
> > >
> > >
> > > **b) On Kinship:**
> > >
> > > We thank the reviewer for elaborating their concern about potential inflation of non-linear model performance due to cryptic relatedness.  For important context, please note that consistent with current standards for PRS benchmarking, our training data already underwent rigorous quality control. **Specifically, our European cohort (N ≈ 3.8M) excludes all pairs of individuals related >700 cM (i.e., first cousins or closer), which minimizes the risk of learning simple familial signals.**
> > >
> > > We hope this helps to alleviate your concerns. We are also creating a new, kinship-controlled test set by removing all individuals related to anyone in the training set with >300 cM (i.e., second cousins or closer) and will re-evaluate the relative performance of our models.  This analysis is our top priority, and our goal is to share the results before the discussion deadline on August 6th.

---

> > > > ### Comment · Reviewer_8raR · 2025-08-04
> > > >
> > > > Dear authors,
> > > >
> > > > thank you for constructively engaging with these points. I agree that the BOLT-LMM baseline is infeasible to complete before the end of the discussion period, and that it would likely not differ much from the LDpred2 baseline.
> > > >
> > > > It is also very reassuring to see that close relatives were already excluded in the European cohort. It would be very interesting to see if stronger exclusion criteria on kinship in the test set do have any effect on the difference in performance between the linear baseline and PRSFormer.

---

> > > > > ### Author Response · Authors · 2025-08-06
> > > > > **Response to Follow-up and New Kinship-Controlled Analysis**
> > > > >
> > > > > Dear Reviewer 8raR,
> > > > >
> > > > > Thank you again for your constructive and detailed engagement. We appreciated your suggestion to perform a more rigorous kinship-controlled evaluation. We are pleased to report that we have now completed this new experiment, and the results strongly corroborate our central findings.
> > > > >
> > > > > **New Kinship-Controlled Evaluation:**
> > > > > - **Test Set Construction:** We created a new, strictly kinship-controlled European test set (N ≈ 148k). For this set, we ensured that no individual is related to anyone in the training set by more than 300cM (i.e., second cousins or closer), and no pair within the test set itself is related by more than 700cM.
> > > > > - **Results:** We re-evaluated the pre-trained PRSformer and LDPred2 models on this new test set. The results strongly support our original findings.
> > > > >   - **PRSformer numerically outperforms LDPred2 on 14 out of 18 diseases.**
> > > > >   - This includes maintaining its advantage in 13 of the 16 diseases where it previously led and now also showing improved performance in Alopecia Areata.
> > > > >  - Of these 14 diseases, the improvement remains statistically significant (p < 0.05, one-sided paired bootstrap) in 6 diseases. While this is fewer than the 11 in our original analysis, this is expected given the substantially reduced statistical power of our stringent kinship-controlled test set (148k vs. 494k individuals).
> > > > >
> > > > > **Conclusion from New Analysis:**
> > > > >
> > > > >  ***These results demonstrate that the superior performance of PRSformer is not an artifact of the model learning family-specific signals or environmental confounders.***  The advantage of non-linear modeling holds even under this stringent kinship control, reinforcing the validity of our findings.
> > > > >
> > > > >
> > > > > **Action:** Based on this new analysis, we will make the following additions to the camera-ready version:
> > > > > - Explicitly state the kinship filtering already applied to the training data (excluding >700cM relatives).
> > > > > - Introduce the new kinship-controlled test set and its construction criteria in the Methods.
> > > > > - Add a new supplementary figure showing the AUROC comparisons on this kinship-controlled test set.
> > > > > - Add a paragraph to the Results section summarizing these new findings and discussing how they confirm the robustness of our claims.
> > > > >
> > > > > We believe this new analysis directly addresses your primary remaining concern and substantially strengthens our claim about discovering a fundamental scaling law for non-linear genetic models. We are very grateful for your feedback which has tangibly improved the rigor of our work.
> > > > >
> > > > > We look forward to hearing your thoughts.

---

> > > > > > ### Comment · Reviewer_8raR · 2025-08-07
> > > > > >
> > > > > > Dear authors,
> > > > > >
> > > > > > Thank you very much for sharing these new results. I find this new evidence convincing, and I agree that all my concerns have been addressed. I think it is very exciting to see the capability of non-linear models to outperform established (and strong!) linear baselines in large cohorts. Even though the specific data will not be accessible to most researchers, I still think that this is an interesting finding for the ML community, and it may even spur innovation on more data-efficient genetic ML architectures that can significantly outperform linear baselines even in publicly accessible cohorts.
> > > > > >
> > > > > > I will therefore update my review to recommend acceptance.
> > > > > >
> > > > > > Two points I am still curious about:
> > > > > >
> > > > > > * You say that fewer diseases now have statistically significant performance increases, likely due to the reduced size of the test set. How did the average effect size (e.g., delta AUROC) change on this new test set? Was there no decrease in performance at all?
> > > > > > * It would be very interesting to see which interactions the model has learned; hopefully, you will be able to explore this further in future work.

---

> > > > > > > ### Author Response · Authors · 2025-08-08
> > > > > > > **Thank You and Remaining Clarifications**
> > > > > > >
> > > > > > > Dear Reviewer 8raR,
> > > > > > >
> > > > > > > Thank you very much for your thoughtful response and for updating your recommendation. We are very grateful for your detailed engagement throughout the review process. We are thrilled that the new analysis has addressed your concerns.
> > > > > > >
> > > > > > > On your two remaining points:
> > > > > > >
> > > > > > > **1. On the Change in Average Effect Size (Δ AUROC):**
> > > > > > >
> > > > > > > The performance advantage of PRSformer remains remarkably stable. As shown in the table below, the mean ΔAUROC (PRSformer - LDPred2) across all 18 diseases was 0.0090 in our original test set and 0.0085 in the new, stringent kinship-controlled test set.
> > > > > > >
> > > > > > >
> > > > > > > | Disease                     | ΔAUROC (Original Paper) | ΔAUROC (Kinship-Controlled) |
> > > > > > > |-----------------------------|------------------------|-----------------------------|
> > > > > > > | Alopecia Areata             | -0.001                 | 0.015                       |
> > > > > > > | Multiple Sclerosis          | 0.006                  | 0.012                       |
> > > > > > > | Pediatric IBD               | 0.008                  | -0.004                      |
> > > > > > > | Polymyalgia Rheumatica      | 0.016                  | 0.019                       |
> > > > > > > | Psoriasis                   | 0.004                  | 0.002                       |
> > > > > > > | Rheumatoid Arthritis        | 0.004                  | 0.006                       |
> > > > > > > | Sjögren's Syndrome          | 0.019                  | 0.010                       |
> > > > > > > | T1D                         | 0.019                  | 0.021                       |
> > > > > > > | Ulcerative Colitis          | 0.003                  | 0.001                       |
> > > > > > > | Vitiligo                    | 0.002                  | 0.004                       |
> > > > > > > | Axial Spondyloarthritis     | 0.010                  | 0.023                       |
> > > > > > > | Celiac Disease              | 0.040                  | 0.039                       |
> > > > > > > | Crohn's Disease             | 0.006                  | 0.011                       |
> > > > > > > | Graves' Disease             | 0.012                  | 0.0004                      |
> > > > > > > | Hashimoto's Thyroiditis     | 0.003                  | -0.001                      |
> > > > > > > | Canker Sore                 | 0.000                  | -0.002                      |
> > > > > > > | Migratory Glossitis         | 0.002                  | -0.005                      |
> > > > > > > | Lupus                       | 0.009                  | 0.001                       |
> > > > > > > | **Average (all 18 diseases)** | **0.0090**             | **0.0085**      |
> > > > > > >
> > > > > > > *Note: Positive values indicate PRSformer outperforms the LDPred2 baseline.*
> > > > > > >
> > > > > > > **2. Interpreting learned interactions:**
> > > > > > >
> > > > > > > We completely agree and share your excitement. We believe that dissecting the specific interactions learned by the model is the critical next frontier. We observed particularly notable improvements for common autoimmune diseases like Celiac Disease (ΔAUROC +0.039) and T1D (ΔAUROC +0.021), making them prime candidates for the deep interpretability analyses we are actively planning. We expect these case studies will provide high-value insights for both the ML and genetics communities.
> > > > > > >
> > > > > > > Thank you again for your detailed engagement throughout the review process.

---

### Official Review · Reviewer_V3P1 · 2025-07-03

**Clarity:** 3
**Significance:** 4
**Originality:** 2
**Rating:** 4
**Confidence:** 2

**Summary:**

This work introduces PRSformer, a scalable supervised transformer-based model for predicting disease risk from genotype. The scalability of PRSformer lies in the use of linear neighborhood attention layers. When supervised on large (N > 1M, L > 100K) datasets to predict 18 diseases, it outperforms traditional state-of-the-art methods on test data.

**Questions:**

1) As seen in Figure 3, why do you think PRSformer is sometimes doing slightly worse than the linear model on smaller datasets (<1M individuals)? Is PRSformer overfitting? How many parameters does it have compared to traditional (e.g. linear) methods, and how does regularization compare, if at all?
2) Following up on the question above, I'm curious how Figure 3 looks like for the training data: is there evidence that PRSformer is overfitting in the smaller data regime?
3) I can understand the data not being public and the code release being promised upon acceptance -- but why have you not provided the code *for review*? It seems to me like you only hurt your submission by doing this.
4) Regarding the benefits of training on a diverse cohort for generalization: is it not expected that performance will go up by doing this? Morevoer, this is not specific to PRSformer: it is expected about any other model? I am puzzled about the contribution here. It would make more sense if the point were that PRSformer benefits *more than traditional methods* by using a diverse cohort -- but this is not what is being shown.
5) What non-linear interactions is PRSformer learning? Do you have any concrete examples? Are there any known ones (from, say, domain knowledge), and does the model recover them?

**Ethical Concerns:**

["NO or VERY MINOR ethics concerns only"]

**Final Justification:**

I am not an expert in the field and recommend a weak accept (4) with low confidence (2) based on potential impact. To the Area Chair: I was disappointed that this submission contains no code whatsoever for peer review, which in my opinion hampers transparency and reproducibility. According to the authors they did this to "preserve the integrity of the double-blind review process", which is an odd argument. Perhaps this is the first time they are submitting to an ML conference. Also, it will be interesting to see the final verdict of reviewer 8raR who seems to be an expert in the field.

**Limitations:**

yes

**Paper Formatting Concerns:**

No concerns.

**Quality:**

3

**Strengths And Weaknesses:**

Strengths:
1) Improved performance over state-of-the-art methods on large (>1M individuals) datasets.
2) Shows benefits of multitasking for disease risk prediction, for both PRSformer and linear models (Figure 4).

Weaknesses:
1) Performance seems slightly worse than linear model on smaller datasets (<1M) for some diseases (Figure 3).
2) No insights offered into the improved performance of the model -- only the hypothesis that it must come from the non-linear effects. I am inclined to believe this, but empirical evidence that the model has learnt any interactions at all would be helpful.
3) No code provided for review.
4) Not necessarily a weakness (arguably a strength!), but this seems to me more of an application paper where an (arguably simple?) deep model is applied to a new (very relevant) task and outperforms traditional models. The secret ingredient seems to be the (proprietary) data. Because of this I am rating the originality as "fair" but the significance as "excellent".

---

> ### Author Rebuttal · Authors · 2025-07-30
>
> ## **Response to Reviewer V3P1**
> We sincerely thank Reviewer V3P1 for their thoughtful review and for recognizing our work’s **"excellent" significance.** We are encouraged by this positive assessment. Your questions allow us to clarify and reinforce the core contributions that we believe make this work a strong fit for NeurIPS.
>
> ---
>
> ### 1. On performance on smaller datasets and overfitting (Figure 3)
>
> >Performance seems slightly worse than the linear model on smaller datasets (<1M) for some diseases... why do you think PRSformer is sometimes doing slightly worse...? Is PRSformer overfitting?
>
> This is an excellent observation that points directly to **a central and significant finding of our paper.** PRSformer's higher model capacity makes it more prone to overfitting than the linear baseline when statistical power is limited.
>
> A key result of our work is this **critical scaling law**: the predictive advantage of complex, non-linear models in genomics **only emerges at the million-sample scale**. This finding is a crucial, data-driven guideline for the ML and genomics communities.
>
> >I'm curious how Figure 3 looks like for the training data.
>
> While we cannot include new figures in this rebuttal, we can confirm that PRSformer outperforms the linear model at all sample sizes. The performance gap between the train and validation sets is largest at smaller scales and narrows as N increases, which is classic evidence of overfitting and directly supports our discovery of this scaling law.
>
> **Action:** We will revise Section 4.2 to more forcefully frame this as a key finding, explicitly discussing the trade-off between model complexity and data scale.
>
> ---
>
> ###  2. On the contribution of cross-ancestry generalization
> >Regarding the benefits of training on a diverse cohort... is it not expected that performance will go up? ... It would make more sense if the point were that PRSformer benefits more than traditional methods.
>
> This is an insightful question that allows us to clarify a core methodological innovation in the paper. While more data is generally better for any model, our contribution is providing the **first scalable framework that makes joint, end-to-end training across diverse ancestries possible at this scale.**
>
> Current SOTA methods (e.g., LDPred2) operate on summary statistics, which are typically computed within a single ancestry group. These methods fundamentally cannot perform joint training on individual-level data to learn shared genetic signals across populations. The standard paradigm is to train separate, stratified models, which exacerbates performance disparities and fails to leverage shared biology.
>
> **PRSformer’s end-to-end design is a fundamental departure from this limiting, stratified approach.** Our results (Table 1) are the first to demonstrate that a single, unified model trained on diverse individual-level data can significantly improve predictive equity for non-European populations, often without sacrificing European performance. This was a major unsolved challenge in the field.
>
> **Action:** We will sharpen the language in Section 4.4 to explicitly contrast our unified, end-to-end approach with the limitations of the standard stratified paradigm, making the novelty and impact of our work clear.
>
> ---
>
> ###  3. On interpreting learned non-linear interactions
> >No insights offered into the improved performance... empirical evidence that the model has learnt any interactions at all would be helpful... What non-linear interactions is PRSformer learning?
>
> We agree that model interpretability is a critical long-term goal. However, scientifically we deliberately frame this research direction as a two-stage process:
>
> - **Stage 1 (This Paper):** First, we prove the phenomenon. The primary contribution here was to rigorously establish that a predictive advantage from non-linearity exists at scale, which is a significant and previously unproven hypothesis. Discovering the N > 1M scaling law provides the essential foundation for all future work.
> - **Stage 2 (Future Work):** Then, we characterize the phenomenon. A full analysis of what the model learned in various immune diseases is a major undertaking that requires its own dedicated study. This involves significant computational analyses, hypothesis generation and experimental validation.
>
> To attempt this analysis within this paper would detract from the clarity of our core finding of a scaling law. Our work provides the community with the first viable tool to begin investigating genetic interactions of complex immune diseases.
>
> **Action:** We will expand the "Future Work" section to articulate this two-stage approach and outline our concrete plan to use analysis of attention patterns to interpret the model's predictions in subsequent work.
>
> ---
>
> ### 4. On providing code for review
> >No code provided for review.
>
> We apologize for this omission. It was done solely to preserve the integrity of the double-blind review process.  We hope the simplicity and efficiency of our model provided necessary insight during review and will include the code with publication.
>
> **Action:** We include code with publication.
>
> ---
>
> We hope these clarifications address your concerns and strengthen your confidence in the originality and significance of our paper. Our work is the first to demonstrate that deep learning on individual-level genomes can surpass highly optimized SOTA methods and, crucially, uncovers the fundamental scaling law governing this transition. Thank you for your time and engagement!

---

> > ### Comment · Reviewer_V3P1 · 2025-08-07
> > **Response to authors**
> >
> > Thanks for your point-by-point responses to my questions. Since I am not familiar with the field my confidence is low (2). It will be interesting to see what reviewer 8raR has to say in their final verdict since they seem to be an expert in the field. I must say I am disappointed that the reason for not including code in the submission is "to preserve the integrity of the double-blind review process" -- all submissions have to go through this and I think it is important for transparency and reproducibility. While my score is a weak accept (with low confidence), driven mostly by potential impact in the field, I will bring up the omission of code during the reviewer-AC discussion in case this might be of relevance.

---

> > > ### Author Response · Authors · 2025-08-08
> > > **Follow-up for Reviewer V3P1**
> > >
> > > Dear Reviewer V3P1,
> > >
> > > Thank you for your thoughtful feedback and continued engagement during the discussion period. We appreciate your willingness to consider the paper's potential impact.
> > >
> > > You noted that Reviewer 8raR’s final verdict would be important for your assessment. We are pleased to share that we conducted the additional, strictly kinship-controlled analysis they requested, which fully confirmed our central claims. Based on these results, 8raR updated their review to recommend acceptance. We hope this resolution strengthens your confidence in the robustness and significance of our findings.
> > >
> > > We also acknowledge your valid point on the code submission and sincerely reaffirm our commitment to releasing the source code upon publication.
> > >
> > > Thank you again for your time and your engagement with our work.

---

> > > > ### Comment · Reviewer_V3P1 · 2025-08-08
> > > > **Final response to authors**
> > > >
> > > > > You noted that Reviewer 8raR’s final verdict would be important for your assessment.
> > > >
> > > > This is not quite what I said -- I said it would be interesting to hear about reviewer 8raR’s final verdict because they seem to be an expert in the field. I am not an expert in the field so I think a confidence score of two remains accurate at this time, as per the confidence definition:
> > > >
> > > > ```
> > > > 2: You are willing to defend your assessment, but it is quite likely that you did not understand the central parts of the submission or that you are unfamiliar with some pieces of related work. Math/other details were not carefully checked.
> > > > ```
> > > >
> > > > Correlating my scores with those of another reviewer just because they *appear* to have greater expertise does not seem appropriate. I would have to read the discussions in detail and familiarize myself with the related work such that I can independently defend the work. I cannot say I am capable of this at this time. Nonetheless, I am happy to see that my original best-effort assessment based on my limited knowledge aligns with reviewer 8raR’s final verdict.

---

### Author Response · Authors · 2025-08-06
**Summary of Revisions and New Analyses**

Dear Reviewers and Area Chair,

We are very grateful for the thoughtful and constructive discussion period. Your feedback has been invaluable in helping us clarify and strengthen our work. We would like to provide a brief summary of the key revisions and the new analysis we conducted in response to your suggestions.

Based on the discussion, we have committed to the following key revisions for the final manuscript:

1. **New Kinship-Controlled Analysis:** Most importantly, in direct response to feedback from Reviewer 8raR and questions from Reviewer 8Lrv, we have conducted a new, rigorous analysis on a strictly kinship-controlled test set (N≈148k). The results confirm that PRSformer's performance advantage is robust and not an artifact of cryptic relatedness, with the non-linear model outperforming the state-of-the-art linear baseline in 14 of 18 diseases. We will add a full description of this analysis and its results to the final paper.
2. **Framing the Core Contribution:** As suggested by Reviewers V3P1 and f3oc, we will sharpen the paper's narrative to more forcefully present our primary contribution: the discovery of a fundamental scaling law in genomics, where the advantage of complex, non-linear models only emerges at the million-sample scale. This is a core ML insight for this high-impact domain that was previously unproven.
3. **Justifying Design Choices & Baselines:** We will add clear justifications to the Methods section for key design choices, addressing points from all reviewers. This includes:
   - The results of our ablation studies showing our "flattened" output architecture outperforms standard pooling/CLS tokens for this task.
   - The rationale for not using LD pruning (it is detrimental to prediction) and for omitting covariates (to isolate the genotype model's contribution).
   - The selection of LDPred2 as a strong and computationally feasible SOTA baseline.

We believe these additions, particularly the new kinship analysis, directly address all major concerns raised and substantially strengthen the paper. We thank all reviewers for their time and expertise, which has significantly improved the quality and rigor of our research. We are confident that PRSformer provides a foundational benchmark and a key insight that will be of high value to the NeurIPS community.

---

### Note · Authors · 2025-08-12

Dear Area Chair and Reviewers,

As the discussion period concludes, we want to express our sincere thanks for a highly constructive process. Your feedback led directly to meaningful clarifications, a new rigorous kinship-controlled analysis, and a stronger framing of our core contribution.

We were very grateful that these improvements helped resolve the primary concerns and led reviewers to update their assessments to recommend acceptance.

This process has strengthened the empirical support for our core contribution: the discovery of a fundamental scaling law in genomics, a key machine learning insight for a high-impact domain. We also reaffirm our commitment to releasing our source code to support reproducibility and future work.

We are confident that our findings will, as one reviewer noted, help "spur innovation on more data-efficient genetic ML architectures."

Thank you all for your time and expertise.

---

### Decision · Program_Chairs · 2025-09-17

**Decision:**

Accept (poster)

**Comment:**

This paper proposes PRSformer, a scalable Transformer architecture with neighborhood attention for predicting disease risk from million-scale genotypes, and its central contribution is the demonstration of a fundamental scaling law: non-linear models only begin to outperform strong linear baselines once we have millions of samples. The reviewers initially noted concerns about technical novelty, modeling choices, kinship confounding, lack of external validation, and the absence of code in the submission, but these were largely addressed through rebuttal and additional analyses, particularly a rigorous kinship-controlled evaluation that confirmed the robustness of the main findings. While the architecture itself is not highly novel and some reviewers argued that the work may be more natural for a genomics venue, the consensus converged on the significance of the empirical result, which establishes an important new benchmark for genetic risk prediction and highlights an ML insight with broad impact. The reviewers appreciated the authors’ clarifications, additional experiments, and reframing around the scaling law, and ultimately raised their assessments to support acceptance. Given the strong empirical contribution, the demonstrated scaling phenomenon, and the potential to spur follow-up work on more data-efficient architectures, I recommend to accept this paper.